# REINFORCEMENT LEARNING FOR ADMISSION CONTROL IN TWO-SIDED QUEUEING SYSTEMS

## ABSTRACT

Two-sided queues are a useful formalism for modeling two-sided markets, as well as more general systems in which work is conserved. Furthermore, in practical applications the arrival rate of different entities is often unknown, and may vary based on the state. General-purpose reinforcement learning algorithms may struggle at scale due to the dependency on the diameter of the Markov Decision Process (MDP), which often scales exponentially over the state space in queueing systems. To solve these issues, we present an algorithm with a diameter-independent regret bound, for the problem of admission control in a state-dependent two-sided queue. Where $S$ is the size of the state space, $N$ is the number of types, $T$ is the number of steps and $\kappa$ is the ratio between the upper and lower rate bounds, our algorithm can be shown to have a regret bound of $\tilde{O}(\kappa^3 S^{1.5}\sqrt{T} + \kappa^{2.5} S^{1.5}\sqrt{NT})$. We then show that this can significantly outperform general-purpose algorithms in an empirical study.

## 1 INTRODUCTION

Admission control problems, in which an agent decides whether or not to admit different entities into a queue, have found widespread application within the fields of mobility (Afèche et al., 2023; Wang et al., 2024), communications (Ahmed, 2000; Ghaderi & Boutaba, 2006), inventory systems (Ionnidis, 2011), distributed computing (Park & Humphrey, 2010) and online marketplaces (Su & Li, 2024). Two-sided queues are of particular interest in admission control problems (Su & Li, 2024; Liu & Weerasinghe, 2022; Doval & Szentes, 2025) as they can be used to model more general situations in which the agent may act as a middleman between complementary entities. In a two-sided queue, each entity in the system is either a customer or a server. Customers and servers, either of which may wait, must be paired to depart the system. For example, ride-hailing systems and online task marketplaces can be modeled as two-sided queues, since providers may wait for customers and vice versa.

Model-based reinforcement learning algorithms, such as the landmark UCRL2 algorithm (Auer et al., 2008), are attractive candidates for queue control problems, including admission control. However, any general purpose reinforcement learning algorithm is known to have a regret bound of at least $O(\sqrt{DSAT})$ (Auer et al., 2008), where $D$ is the diameter of the MDP, $S$ is the size of the state space, $A$ is the size of the action space, and $T$ is the number of steps. This is particularly challenging since problems involving even the most elementary queueing systems, including the $M/M/1/k$ queue, can have a diameter that varies exponentially over $S$ (Anselmi et al., 2022; Weber et al., 2024). Therefore, there is a significant risk of poor performance at moderate state counts and above.

A recent stream of papers (Anselmi et al., 2022; Weber et al., 2024; Anselmi et al., 2024; Dai & Gluzman, 2022; Jali et al., 2024; Staffolani et al., 2023) aims to mitigate this problem by developing problem-specific reinforcement-learning algorithms for the control of queueing systems. In (Weber et al., 2024), the authors consider an admission control problem in a single queue with a constant arrival rate and known service rate. Using a bound on the bias, as well as problem-specific simplifications of the learning algorithm, they develop an algorithm based on UCRL2 with a dominant regret term that is independent of the state size for queues in which there is a known service rate and constant arrival rate. Similarly, (Anselmi et al., 2022) considers the problem of service rate control, with a dominant regret term that varies with the weighted second moment of a reference policy,

but not with the diameter or number of states. Some papers include countable state-space models (Dai & Gluzman, 2022; Comte et al., 2025; Yang et al., 2025) with state-space independent bounds. However, these generally use policy-gradient or deep reinforcement learning methods and do not achieve square-root minimax regret bounds. Furthermore, these generally require a small parameter space, which may cause problems with scalability. For example, in (Comte et al., 2025), there is an application to admission control in a finite-capacity M/M/1 queue. It is shown that the regret scales exponentially with the state space in this application due to a dependence on a Lyapunov drift parameter. (Yang et al., 2025) enables queue control in an infinite-capacity two-sided queue, but the queue is allowed to grow to infinity with time to enable a tractable fluid solution to be optimal.

However, much of the prior work on learning in queueing systems, including admission control problems (Weber et al., 2024; Anselmi et al., 2024), and others (Anselmi et al., 2022; Liu et al., 2019; Raeis et al., 2021), rely in part on strong assumptions, such as homogeneity in system parameters across different states. (Anselmi et al., 2024) allows for limited state-dependence under the assumption that the queue is a flow-equivalent server of a Jackson network, but does not enable more general state-dependence of rates. In practical situations, arrival and service rates may be state-dependent, due to strategic behavior and practical limitations on the waiting buffer (Hassin, 2016; Naor, 1969; Bekker, 2005). Therefore, a natural extension is to see if a diameter-independent upper-confidence reinforcement learning algorithm is possible with more general state-dependent arrival and service rates. In particular, learning state-dependent rates requires greater exploration over the state space. Since the diameter is exponential over the state space, exploration is particularly challenging in queueing systems. However, in this paper we establish that attractive regret bounds are still achievable, subject to a monotonicity assumption.

## 1.1 CONTRIBUTIONS

In this work, we establish a learning algorithm in which the regret bound is independent of the diameter and has polynomial constants over the size of the state space, in contrast to the exponential bounds from general-purpose algorithms. In fact, the regret scales according to $S^{3/2}$ in the dominant (square-root) term, and $S^3$ in the non-dominant (logarithmic) term. This algorithm applies to situations in which arrival rates may be state-dependent, under a simple monotonicity assumption. This implies that the queueing systems are learnable at scale, even when the system cannot be parametrized by a small number of states. In contrast to prior work, the arrival and service rates are allowed to depend on the current state, with a mild monotonicity assumption, while maintaining attractive regret guarantees. This algorithm applies to both two-sided queues as well as more traditional one-sided ones. To maximize the generality of the model, we allow arriving entities to offer both positive as well as negative rewards, and we allow positive holding rewards as well.

## 1.2 PAPER STRUCTURE

The rest of the paper is organized as follows. Section 2 describes the two-sided queueing model as well as the formulation of the admission control problem. Section 3 describes the proposed UCRL-TSAC algorithm for admission control in a two-sided queue. Section 4 presents the regret bound as well as supplementary results. Section 5 gives results on the empirical performance, alongside general-purpose baselines from the literature. Detailed proofs of the results, as well as supplementary material on the algorithm, are covered in the Appendix.

## 2 PROBLEM FORMULATION

We consider the problem of admission control in a two-sided queue. In a two-sided queue, entities are either classified as customers or servers. An agent decides whether to admit or reject an arriving entity. If there is at least one server in the system, then any admitted customers will match with a server and both will leave the system. If there are no servers available, an admitted customer will join the customer side of a queue and wait for a server to arrive. Likewise, any arriving servers will pair with a customer and leave the system if any customers are present in the system. Otherwise, it will join the server side of the queue and wait for an arriving customer. There are no compatibility requirements, and an arriving server can accept into service any arriving customer irrespective of its type, and likewise for arriving customers when there are waiting servers in the system.

Since decisions are made frequently (Puterman, 2005), we maximize the long-run average reward, also known as the gain. Rewards are gained upon the admittance of an entity, the abandonment of an entity, and per unit time. These rewards are also dependent on the particular state. Rewards may be negative, but are assumed to have known bounds.

We use a similar problem formulation to (Miller, 1969; Su & Li, 2024; Feinberg & Yang, 2011; Weber et al., 2024; Feinberg & Reiman, 1994), by considering a single-class system with reward differentiation between different customer and server types. The state can be represented as an integer. When the state is equal to 0, no customers or servers are present in the system. If the state is equal to $s > 0$, there are $s$ customers present and no servers. If the state is equal to $s < 0$, then there are $-s$ servers present and no customers. Each side of the queue is finite-capacity, with $\bar{s}$ being the capacity for the customer side and $-\underline{s}$ being the capacity for the server side. The total number of states is equal to $S = \bar{s} - \underline{s} + 1$. Customers and servers are scheduled in first-come first-serve (FCFS) order. In contrast to much of the reinforcement learning literature, but similarly to much of queueing theory, the model is continuous-time.

There are $N$ total customer types, distinguished between $N^c$ different customer types and $N^s$ different server types. Servers and customers arrive with exponentially-distributed and state-dependent inter-arrival times. We represent the arrival rate for type-$i$ customers, in state $s$, as $\lambda_i(s)$. The arrival rate for type-$j$ servers is given as $\mu_j(s)$. In state $s > 0$, a customer abandonment occurs according to an exponentially-distributed random variable with rate $\gamma(s)$. Similarly, in state $s < 0$, a server abandonment occurs according to an exponentially-distributed random variable with rate $\eta(s)$. For notational convenience, we use $\gamma(s) = 0$ for all $s \leq 0$, and $\eta(s) = 0$ for all $s \geq 0$. The total customer arrival rate is represented as $\Lambda(s) = \sum_i^{N^c} \lambda_i(s)$, and the total server arrival rate is represented as $M(s) = \sum_j^{N^s} \mu_j(s)$. Although we use the term abandonment in the paper, this may be used to model other service processes outside the arrival of complementary entities. This may include traditional service processes or removal from the queue. Therefore, this model generalizes service processes.

We assume there is a known lower bound for event rates, $q^{\min} > 0$. For $s > 0$, we assume that $\gamma(s) > q^{\min}$ and for $s < 0$, $\eta(s) > q^{\min}$. In all states, let $\Lambda(s) + \eta(s) > q^{\min}$ and $M(s) + \gamma(s) > q^{\min}$. Furthermore, we assume that there are known upper bounds for the total arrival and service rates across all states, $\lambda^{\max}$ and $\mu^{\max}$, respectively. Likewise, there are known upper bounds for abandonment rates, $\gamma^{\max}$ and $\eta^{\max}$. We denote the maximal rate as $q^{\max} = \max(\gamma^{\max}, \eta^{\max}) + \lambda^{\max} + \mu^{\max}$. The ratio between the maximal and minimal rates is denoted by $\kappa = \frac{q^{\max}}{q^{\min}}$. We note that the rate bounds are state-independent.

Upon accepting a type-$i$ customer arrival in state $s$, the agent receives a reward of $r_i^c(s)$. A reward of $r_j^s(s)$, is similarly received upon accepting a type-$j$ server arrival in state $s$. Upon an abandonment in state $s$, a reward of $r^a(s)$ is received by the agent. There is also a state-dependent reward, $r^h(s)$, which is received per unit of time that the system spends in state $s$. All reward values are assumed to be bounded in the interval $[-1, 1]$.

Upon state changes, the agent decides on an action $a \in \mathcal{A} = 2^{\{1...N^c\}} \times 2^{\{1...N^s\}}$. This action includes the subset $a_1$ of customer types to admit, as well as the subset of server types $a_2$. Under action $a$ in state $s$, the state transitions to $s + 1$ with rate $\sum_{i \in a_1} \lambda_i(s) + \eta(s)$, and to state $s - 1$ with rate $\sum_{j \in a_2} \mu_j(s) + \gamma(s)$. No other transitions are possible, and therefore this is a birth-death process. We use $\lambda_\pi(s) = \sum_{i \in a_1} \lambda_i(s)$ to represent the arrival rate of accepted customers at state $s$ under policy $\pi$. Likewise, we use $\mu_\pi(s) = \sum_{j \in a_2} \mu_j(s)$ to represent the arrival rate of accepted servers at state $s$ under policy $\pi$.

Without loss of generality, the optimal policy can be found considering only the restricted action set $\tilde{\mathcal{A}}$. Let $\geq_l$ be a lexicographical ordering. This is defined as the set of all actions $a$ such that

$$a_1 \in \{\emptyset\} \cup \{\{l \in [1, N^c] | (r_l^c, l) \geq_l (r_i^c, i)\} \quad \forall i \in [1, N^c]\}$$
$$a_2 \in \{\emptyset\} \cup \{\{l \in [1, N^s] | (r_l^s, l) \geq_l (r_j^s, j)\} \quad \forall j \in [1, N^s]\}$$

In particular, the restricted action set contains all actions such that either no customer (server) types are accepted, or there exists a particular threshold type $i$ ($j$), for a given action, such that all types

with a greater reward are accepted. The use of a lexicographical ordering with respect to the index is arbitrary, but it is necessary to distinguish between different types with identical rewards. A full proof of the optimality of this action set can be found in the Appendix, specifically Lemma 1 and Corollary 1. The use of the restricted action set is needed for computational tractability, as it reduces the number of possible actions from $2^{N^c + N^s}$ to $(N^c + 1)(N^s + 1)$.

The average reward under action $a$, $\bar{r}(s, a)$, is equal to

$$\bar{r}(s, a) = \sum_{i=1}^{N^c} 1_{i \in a_1} r_i^c(s) \lambda_i(s) + \sum_{j=1}^{N^s} 1_{j \in a_2} r_j^s(s) \mu_j(s) + r^a(s)\gamma(s) + r^a\eta(s) + r^h(s)$$

We also use $r_\pi(s)$ to represent the average reward in state $s$ under policy $\pi$. Let $g^*$ be the maximal gain, $r_t$ be the reward per unit time in step $t$, $\tau_t$ be the time spent in step $t$, and $k$ be the episode step $t$ is falls in. We define the regret as

$$\Delta_{k,t} = \tau_t(g^* - r_t)$$

Given a policy $\pi$, we represent the total admittance rate of customers, while in state $s$, as $\lambda_\pi(s) = \sum_{a \in \mathcal{A}} \pi(s, a) \sum_{i=1}^{N^c} 1_{i \in a_1} \lambda_i(s)$. The total admittance rate of servers, while in state $s$, is represented as $\mu_\pi(s) = \sum_{a \in \mathcal{A}} \pi(s, a) \sum_{j=1}^{N^s} 1_{j \in a_2} \mu_j$.

As is proven in Proposition 4 in Section 4, there needs to be some assumption on the model to guarantee polynomial bounds on the bias, and therefore the regret. We fulfill this by presenting an assumption on rate monotonicity. In particular, we assume that customer arrivals are decreasing over the state space, while server arrivals are increasing. On the other hand, server abandonments decrease while customer abandonments increase. While this may appear to be limiting, it generalizes multi-server queueing models and applies to several practical uses of state-dependent queues. For example, strategic customers may be more likely to balk or abandon the queue when the waiting time is large (Hassin, 2016; Naor, 1969). This is of particular relevance to two-sided markets in which customers are able to observe the queue, and decide on whether to join based on the waiting time. Furthermore, several applications of state-dependent queues in inventory systems, healthcare and communications feature monotonic rates, with arrival rates decreasing with the number of customers in the system and increasing service rates (Bekker, 2005; Hassin, 2016; Yom-Tov & Chan, 2021). More detail on models that can be reasonably assumed to fulfill this assumption is given in Section D.I of the Appendix.

**Assumption 1.** *The total customer arrival rate, $\Lambda(s) = \sum_{i=1}^{N^c} \lambda_i(s)$, is non-increasing over $s$, while the customer abandonment rate $\gamma(s)$ is non-decreasing over $s$. The total server arrival rate, $M(s) = \sum_{j=1}^{N^s} \mu_j(s)$, is non-decreasing over $s$, and the server abandonment rate $\eta(s)$ is non-increasing.*

It is necessary to distinguish between different models at points. Unless otherwise stated, each individual parameter, such as $\lambda_i(s)$, is assumed to be the parameter of the true system, which is possibly unknown. We distinguish between different models in two ways. For arbitrary models, subscripts are used, and are given after all other subscripts. For example, for an arbitrary model $D$, the type-$i$ customer arrival rate in state $s$ is given as $\lambda_{i,D}(s)$. For the extended model, which is used to find the optimistic policy, we use a superscript for each unknown parameter. For example, we represent the type-$i$ customer arrival rate in state $s$ as $\lambda_i^{ext}(s)$.

For admission control in a one-sided queue, the state-dependent abandonment rate $\gamma(s)$ may be used to represent the rate of service. No server types need to exist for the regret bound to hold, and we can simply consider different types of customers. Therefore, the model presented generalizes several common queueing models including finite-capacity, multi-server Markovian queues.

## 3 Reinforcement Learning Algorithm

In this section, we present the UCRL-TSAC algorithm for admission control. This algorithm is analogous to the UCRL2 (Auer et al., 2008) and CT-UCRL (Gao & Zhou, 2024) algorithms, but it is tailored for the admission control problem and enforces a slightly weaker version of Assumption 1 in the optimistic model. The weaker version is labeled as Assumption 2, and is given in the Appendix.

---

**Algorithm 1** UCRL-TSAC

---

**Require:** Confidence parameter $\delta$
1: Initialize a list of sojourn times, event counts, and visit counts $V_{k-1}(\cdot)$, and event counts $V_{k-1}(\cdot, \cdot)$.
2: **for** $k = 1, 2 \ldots$ **do**
3:     Use Algorithm 2 to find an optimistic policy $\pi_k$.
4:     Set the visit count vector $v_k$ to equal 0 in each state.
5:     **while** True **do**
6:         Observe the new state $s$, update $v_k(s) \leftarrow v_k(s) + 1$
7:         When an entity arrives or an abandonment occurs, record the sojourn time and the event type. If it is an arrival, accept or reject it according to $\pi_k$.
8:         If $v_k(s) \geq V_{k-1}(s)$ for any state $s$, set $V_k(s') = V_{k-1}(s') + v_k(s')$ for all states $s'$, exit and terminate the episode.
9:     **end while**
10: **end for**

---

**Algorithm 2** Policy Finding Algorithm

---

**Require:** Sojourn time observations $\tau_t$, event counts $V_{k-1}(\cdot, \cdot)$, initial confidence parameter $\delta$.
1: Construct the extended model using Algorithm 3.
2: Check the validity of the extended model. If it fails, return $\pi_{k-1}$.
3: Use policy iteration or linear programming to find an optimistic policy $\pi^*$, using the restricted action set of the extended model.
4: Derive a new policy $\hat{\pi}_k$ for the extended model by rejecting customer arrivals in positive transient states and server arrivals in negative transient states.
5: Use the given policy to derive the optimistic policy $\pi_k$ by aggregating actions that only differ by acceptance of the fictitious types $N^c + 1$ and $N^s + 1$.

---

We estimate the probability of different arrivals and abandonments instead of state transitions, since the system may only transition to $s + 1$ or $s - 1$ from state $s$. Furthermore, since a continuous-time process is most appropriate, we need to estimate the event rates in addition to the probabilities. In comparison to the UCRL-AC algorithm (Weber et al., 2024), this does not require known service rates, allows for state-dependent rates, and supports a more general reward reformulation. In comparison to UCRL2 and CT-UCRL, the process of finding an optimistic model is simplified with a priori optimism results.

We use $t$ to represent each step, and $s(t)$ to be the state in which step $t$ is spent. We divide the time into episodes, which are indexed by $k$. At the start of each episode, an optimistic policy $\pi_k$ is found, and it is followed until the end of the episode. We use a similar doubling trick to UCRL2 for episode termination. We terminate the episode when the number of state observations for some state $s$ is equal to the number of observations before episode $k$. An initial confidence parameter $\delta$ must be chosen. We decrease the confidence parameter using a harmonic schedule. We use $\delta_k = \frac{\delta}{V_{k-1}}$ to denote the confidence parameter.

### 3.1 CONFIDENCE INTERVALS

We begin by presenting the corresponding point estimators and confidence intervals used to find an optimistic policy. We introduce the notion of positive and negative events, which will allow us to cleanly enforce Assumption 2. Positive events include customer arrivals and server abandonments, both of which potentially lead, depending on the action, to an increase in the state. Conversely, negative events include server arrivals and customer abandonments. We then estimate the conditional probability of abandonments and arrivals based on whether or not it is a positive or a negative type.

In order to formalize this notion, we introduce an indexing scheme for events. Each arrival $l \in \{1, \ldots, N^c\}$ corresponds to a customer arrival of type $l$, and $l \in \{-N^s, \ldots, -1\}$ corresponds to a server arrival of type $-l$. For notational convenience, we use $l = 0$ to represent an abandonment. Note that no states can experience both customer and server abandonments. Then, let $V_{k-1}(s, l)$ be the number of times an event indexed as $l$ has been observed before episode $k$,

---

**Algorithm 3** Constructing the Extended Model

---

**Require:** Sojourn time observations $\tau_t$, event counts $V_{k-1}(\cdot, \cdot)$, confidence parameter $\delta$

1: For each state $s$, initialize $\gamma^{ext}(s)$ to the lowest possible value respecting the upper bound of the inter-event times $\hat{\tau}_k^+ + \frac{1}{2}\epsilon_k^{\tau^+}(s)$, as well as the lower bound of the conditional event probabilities $\hat{p}_{k,s}^+(0) - \frac{1}{2}\varepsilon_k^{p^+}(s)$.

2: Similarly, for each $s$ initialize $\eta^{ext}(s)$ based on the confidence interval for the inter-event times of negative events, and the corresponding confidence set of conditional probabilities, again using a lower envelope.

3: Update $\gamma^{ext}(\cdot)$ and $\eta^{ext}(\cdot)$ to the lowest possible values that respect Assumption 2 and the bounds $[q^{\min}, \bar{\gamma}]$ and $[q^{\min}, \bar{\eta}]$.

4: For each state $s$, find the maximal values of $\Lambda^{ext}(s) + \eta^{ext}(s)$ and $M^{ext}(s) + \gamma^{ext}(s)$, and initialize accordingly, using the confidence sets for inter-event times.

5: Update $\Lambda^{ext}(\cdot) + \eta^{ext}(\cdot)$ and $M^{ext}(\cdot) + \gamma^{ext}(\cdot)$ to the highest possible values that respect Assumption 2 and the bounds $[q^{\min}, \bar{\Lambda} + \bar{\eta}]$ and $[q^{\min}, \bar{M} + \bar{\gamma}]$.

6: Greedily assign conditional probabilities based on the rewards for each event, respecting the lower bounds for abandonments and the $\ell_1$ bounds for the conditional probabilities. Assign the difference between the greedy bound for abandonments and the lower bound to the fictitious types $N^c + 1$ and $N^s + 1$, for server and customer abandonments, respectively.

---

and $l(t)$ be the event observed at step $t$. Also, we use $V_{k-1}^+(s) = \sum_{m=1_{s\geq 0}}^{N^c} V_{k-1}(s, m)$ and $V_{k-1}^-(s) = \sum_{m=-N^s}^{-1_{s\leq 0}} V_{k-1}(s, m)$ to be the number of observed positive and negative events before episode $k$, respectively. Let $V_{k-1,t}^+(s)$ and $V_{k-1,t}^-(s)$ be the number of positive and negative events, respectively, observed up to time $t$ or the end of episode $k-1$. We then present point estimates for the conditional probabilities, where $\hat{p}_{k,s}^+(l)$ is the estimated probability, as of episode $k$, of observing arrival $l$ in state $s$ conditioned on it being positive. Similarly, $\hat{p}_{k,s}^-(l)$ is the estimated probability of observing event type $l$ conditioned on it being negative. The point estimates are given below.

$$\hat{p}_{k,s}^+(l) = \frac{V_{k-1}(s, l)}{V_{k-1}^+(s)} \qquad\qquad \hat{p}_{k,s}^-(l) = \frac{V_{k-1}(s, l)}{V_{k-1}^-(s)}$$

Confidence sets for the conditional probabilities are found separately for customer and server arrivals. We use an $\ell_1$ bound for each, similarly to UCRL2 (Auer et al., 2008). We use $\varepsilon_k^{p^+}(s)$ and $\varepsilon_k^{p^-}(s)$ for the $\ell_1$ bounds of conditional positive and negative events at state $s$, respectively.

$$\varepsilon_k^{p^+}(s) = \sqrt{\frac{2(N^s + 1)}{V_{k-1}^+(s)} \log\left(\frac{2S}{\delta_k}\right)} \qquad\qquad \varepsilon_k^{p^-}(s) = \sqrt{\frac{2(N^c + 1)}{V_{k-1}^-(s)} \log\left(\frac{2S}{\delta_k}\right)} \qquad (1)$$

Since we use a continuous-time model, we must also estimate the total rate of positive and negative events. Confidence intervals based on Hoeffding bounds are not applicable in this case, since the exponential distribution is not subnormal. However, following (Gao & Zhou, 2024; Weber et al., 2024), we can use the truncated empirical mean of (Bubeck et al., 2013) for estimating the time between subsequent positive events, as well as the time between subsequent negative events. Let $\hat{\tau}^+(s)$ and $\hat{\tau}^-(s)$ represent the estimated inter-event times for positive and negative events, respectively, in state $s$. Let $\tau_t^+$ be the total time spent in state $s(t)$ until step $t$ since the last positive event, and $\tau_t^-$ be the total time spent in state $s(t)$ from the last negative event until step $t$. Point estimates for this quantity using the truncated empirical mean are given below

$$\hat{\tau}_k^+(s) = \frac{1}{V_{k-1}^+(s)} \sum_{t=1}^{V_{k-1}} \tau_t^+ 1\left\{ s(t) = s, 1_{s\leq 0} \leq l(t) \leq N^c, \tau_t^+ \leq \sqrt{\frac{2V_{k-1,t}^+(s)}{(q^{\min})^2 \log(\frac{2S}{\delta_k})}} \right\}$$

$$\hat{\tau}_k^-(s) = \frac{1}{V_{k-1}^-(s)} \sum_{t=1}^{V_{k-1}} \tau_t^- 1\left\{ s(t) = s, -N^s \leq l(t) \leq -1_{s\geq 0}, \tau_t^- \leq \sqrt{\frac{2V_{k-1,t}^-(s)}{(q^{\min})^2 \log(\frac{2S}{\delta_k})}} \right\} \quad (2)$$

For each state $s$, half lengths for the confidence intervals around $\hat{\tau}_k^-(s)$ and $\hat{\tau}_k^-(s)$, respectively, are as follows

$$\varepsilon_k^{\tau^+}(s) = \frac{4}{q^{\min}}\sqrt{\frac{2}{V_{k-1}^+(s)}\log\left(\frac{2S}{\delta_k}\right)} \qquad \varepsilon_k^{\tau^-}(s) = \frac{4}{q^{\min}}\sqrt{\frac{2}{V_{k-1}^-(s)}\log\left(\frac{2S}{\delta_k}\right)} \qquad (3)$$

As a heavy-tailed distribution, obtaining tight confidence intervals for the exponential distribution is inherently difficult. The truncated empirical mean, with its corresponding confidence interval, is an attractive candidate due to its simplicity and computation speed. However, the interval can be quite loose in practice. Cantoni's M-estimator (Cantoni, 2010; Bubeck et al., 2013), could also substitute for the estimator presented in (12) and (3), with much tighter confidence bounds. The main disadvantage is that this requires root-finding over the sum of logarithmic terms for each time step and may not necessarily be computationally practical at scale.

## 3.2 Finding an Optimistic Policy

The UCRL-TSAC algorithm differs from prior work such as UCRL2 (Auer et al., 2008) by solving the extended model exactly, as opposed to using an approximate method such as extended value iteration. Since we use bounds over the bias as opposed to the relative value for the regret proof, this is a necessity to obtain attractive regret bounds. This can be made computationally tractable with a notable simplification of the extended model using a priori results. In particular, it can be established that higher customer and server arrival rates always improve the gain, and a similar result can be shown with a conditional distribution with a higher probability of receiving higher-reward customer and server types. The only question that remains is whether a lower or higher abandonment rate is optimistic.

We solve this problem by using the lowest possible value for the abandonment rates from the box bounds for the inter-arrival time and event probabilities, and adding an additional fictitious customer and server type in the extended model. The arrival rate of each fictitious type is equal to the difference in the upper and lower abandonment bounds. Therefore, the choice of admittance of each fictitious type is equivalent to a choice between the upper and lower bound of the abandonment rate. A pseudocode of the algorithm is given in Algorithm 3, which can be completed in $O(S(N+1))$ time. A complete proof of the optimism of the extended model, as well as a more detailed pseudocode for constructing it, is given in Section B of the Appendix. This algorithm is only guaranteed to return a valid model when the extended model falls within the confidence set. If it fails, it is clear the true model falls outside the confidence set and any policy may be used. This respects the regret bounds since we use the worst case deviation between the gain and reward for out-of-confidence episodes in the proof.

Once the extended model has been constructed, the optimal policy may be found using either policy iteration or linear programming. This policy is then adapted into an appropriate policy in two steps. First, it must be modified to reject any arriving customers in positive transient states as well as any arriving servers in negative transient states, which is necessary for the bias bounds and therefore the regret bounds to hold. This does not violate gain-optimality, as changes in the action taken on these states do not change the gain or the set of recurrent states. Secondly, the optimal policy for the extended model can be transformed into one for the true model by aggregating actions that differ only by the acceptance of the fictitious types $N^c + 1$ and $N^s + 1$. The resulting policy for episode $k$ is represented as $\pi_k$. A pseudocode of the algorithm for finding the optimistic policy is given in Algorithm 2.

Once the optimistic policy has been found, we can follow the optimistic policy $\pi_k$ until the number of steps observed in some state $s$ is equal to $V_{k-1}(s)$. Since all arrivals are observable, we do not need to explore based on observed actions. This doubling trick, analogous to that used in UCRL2, is not necessarily optimal, but enables square root minimax bounds and a logarithmic number of episodes over the number of steps.

We also present two results that enable an efficient policy evaluation and improvement without needing to directly solve the gain-bias equation, once the extended model has been constructed. In particular, the system is product-form and therefore the long-run probability and gain can be solved in linear time. Proposition 1 gives the product-form solution for the long-run probabilities, and Proposition 2 establishes a similar result for the bias.

**Proposition 1.** *Under any policy $\pi$, the probability of being in state $s$ in steady-state is equal to*

$$p_\pi^\infty(s) = \begin{cases} \left(1 + \sum_{s''=1}^{\bar{s}} \prod_{s'=1}^{s''} \frac{\lambda_\pi(s'-1)}{\mu_\pi(s')+\gamma(s)} + \sum_{r=0}^{-\underline{s}} \prod_{s'=s''}^{-1} \frac{\mu_\pi(s+1)}{\lambda_\pi(s)+\eta(s)}\right)^{-1} & s = 0 \\ p_\pi^\infty(0) \prod_{s'=1}^{s} \frac{\lambda_\pi(s'-1)}{\mu_\pi(s')+\gamma(s)} & 0 < s \leq \bar{s} \\ p_\pi^\infty(0) \prod_{s'=s}^{-1} \frac{\mu_\pi(s+1)}{\lambda_\pi(s)+\eta(s)} & \underline{s} \leq s < 0 \end{cases}$$

**Proposition 2.** *At any state $s$ such that $s$ and $s+1$ are recurrent, the relative bias*

$$\Delta h_\pi(s) = h_\pi(s+1) - h_\pi(s)$$

*is equal to*

$$\Delta h_\pi(s) = \frac{1}{p^\infty(s)(\lambda_\pi(s) + \eta(s))} \sum_{s'>s} p_\pi^\infty(s')[\bar{r}_\pi(s') - g_\pi] \tag{4}$$

## 4 REGRET BOUNDS

### 4.1 BOUNDING THE BIAS

The principal challenge of reinforcement learning in queues is that many relevant problems have an exponential diameter over the state space (Weber et al., 2024; Anselmi et al., 2022). For example, Appendix A in (Weber et al., 2024) gives an exponential lower bound for the admission control problem in a single-sided M/M/1/S queue, which is among the simplest queueing models. In the regret proof, the difference in the bias between adjacent states is a multiplier on the dominant square root term, and therefore it is necessary to find a bound for this quantity that is polynomial over $S$. We use $h_\pi(s)$ to denote the bias at state $s$ under policy $\pi$. We extensively use the relative bias, $\Delta h_\pi(s) = h_\pi(s+1) - h_\pi(s)$, similarly to (Weber et al., 2024).

We can, perhaps surprisingly, achieve linear bounds over the state space for the bias when Assumption 2 holds. The following proposition establishes this.

**Proposition 3.** *Let Assumption 2 hold. Then, consider a gain-optimal policy $\pi^*$ in which no customer arrivals are accepted in positive transient states, and no server arrivals are accepted in negative ones. Then, for any state $\underline{s} \leq s < \bar{s}$*

$$|\Delta h_{\pi^*}(s)| \leq \frac{2(q^{\max}+1)}{q^{\min}} S$$

In the regret bound, we use $(\Delta h)^{\max} = \frac{2(q^{\max}+1)}{q^{\min}} S$ to represent the upper bound of the relative bias. The proof of bias bounds depends on the time-reversibility of the underlying queueing system, along with the unique aspects of the admission control problem. In particular, using the gain-bias equations of the MDP we can derive a simple rule to tell if an entity can be submitted, namely if the reward is greater than the loss in bias. This gives constant bounds in the case that some types of customers are accepted and others are rejected, or some types of servers are accepted and others rejected. The remaining cases are those in which all customers are rejected and all servers accepted, or vice versa. Then, linear bounds can be found in the remaining cases by an induction argument over the gain-bias equations, and from an argument deriving from the time-reversibility of the system over the recurrent class of states.

One would hope that similar bounds can be found with a more general model. However, it can also be established that in the general case no bound on the relative bias with polynomial bounds can be established, even with bounded rates.

**Proposition 4.** *There exist no polynomial bounds for the bias under an optimal policy, over the number of states, for the set of all possible models under any given rate bounds $q^{\min}, \Lambda^{\max}, M^{\max}, \eta^{\max}, \gamma^{\max}$.*

In the regret proof, the bias of the extended model is used rather than that of the true model. Given that exponential scaling can exist under arbitrary rate bounds, it becomes clear that it is necessary to make some restriction on the set of possible parameters for the extended model. To that end, we explicitly enforce an assumption similar to Assumption 1 by truncating the confidence set.

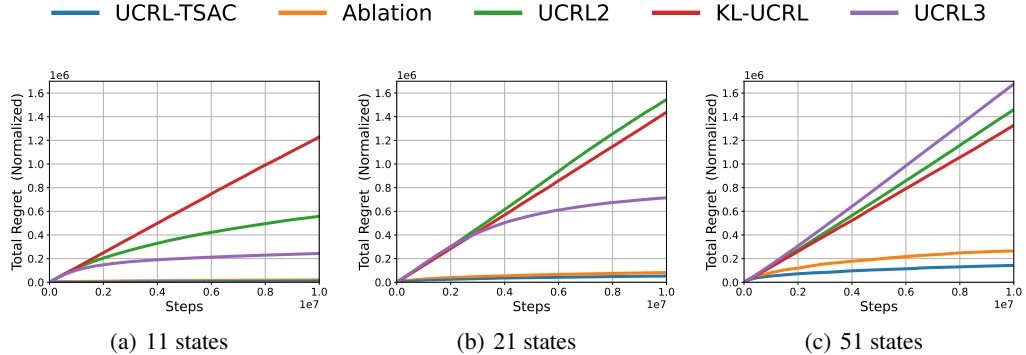

Figure 1: Regret over time at varying state counts, with baselines

## 4.2 REGRET ANALYSIS

Next, we present the core regret proof of the paper. Where $T$ is the time horizon, the result given is typical in format for minimax regret bounds, with a dominant term that varies with $\sqrt{T \log(T)}$ and a non-dominant term that varies with $\log(T)$.

**Proposition 5.** *The total expected regret up to time $T$ is upper bounded*

$$\mathbb{E}\left[\sum_{t=1}^{T} \Delta_{K(t),t}\right] \leq (\sqrt{2}+1)\left[\left(32\kappa^2 + 4\kappa^{1.5}\sqrt{N+1}\right)\sqrt{ST\log\left(\frac{2ST}{\delta}\right)}((\Delta h)^{\max}+1)\right]$$

$$+ \max\left(S, S\log_2\left(\frac{8T}{S}\right)\right)\left[S(\Delta h)^{\max} + (8\delta + 8S\kappa^2)\frac{q^{\max}+1}{q^{\min}}\right]$$

*This corresponds to $\tilde{O}(\kappa^3 S^{1.5}\sqrt{T} + \kappa^{2.5}S^{1.5}\sqrt{NT})$ log-adjusted complexity bounds.*

With regards to the regret bound, the main bottleneck to scaling now becomes the $\kappa^3$ term. This may be prohibitive in situations in which there is significant variability in terms of the rates. The source of this term comes from three points, the bias bound $(\Delta h)^{\max}$, the sojourn time and reward terms in the regret, and finally the confidence bounds on the rate. The first two are intrinsic in the worst case, but the bias may be significantly better empirically in the case that there is a sufficient diversity of arrival types. It remains an open question if the confidence bounds for the inter-event times can be transformed into bounds for the bias without scaling by $\kappa$, or a closely related factor.

The minimax regret bound presented in Proposition 5 is given in terms of the expected regret, rather than high probability bounds as found in (Auer et al., 2008). The use of expected regret bounds is in line with similar papers on reinforcement learning for queue control (Anselmi et al., 2022; Weber et al., 2024). However, we conjecture that high-probability bounds with similar constants may be found by applying appropriate inequalities to the martingale sequences in the regret proof. It should be noted that many common inequalities within the family of Bernstein inequalities are inadequate since the martingale difference sequences with exponentially-distributed terms are neither bounded nor subnormal.

## 5 EMPIRICAL RESULTS

In this section, we present empirical results of our algorithm, compared to an ablation and baselines. For each state count, we use 50 randomly generated models, with an equal capacity for both the server and customer side. We use 3 server and 3 customer types. Total customer and server arrival rates are first generated uniformly between 1 and 5, and are then sorted to fulfill Assumption 1. Individual arrival rates for each type are generated using a Dirichlet distribution with $\alpha = 1$, and are then multiplied by the total customer or server arrival rate. Abandonment rates, for both customers and servers, are generated uniformly between 1 and 1.5 and then sorted as well. State-dependent arrival rewards are generated uniformly within $[-1, 1]$, while abandonment rewards are generated uniformly within the interval $[-1, 0]$. The holding reward is set to $-0.05|s|$ for each state $s$.

Table 1: Average reward ratio

| Algorithm | State Count | | |
|---|---|---|---|
| | 11 states | 21 states | 51 states |
| UCRL2 | 15.8% | 2.4% | 11.1% |
| UCRL3 | 83.4% | 51.2% | $-18.0\%$ |
| KL-UCRL | 61.6% | $-7.2\%$ | 0.1% |
| Ablation | 98.7% | 94.4% | 81.8% |
| **UCRL-TSAC** | **99.0**% | **96.4**% | **90.2**% |

We compare the proposed URCL-TSAC algorithm with three general-purpose baselines from the literature, UCRL2 (Auer et al., 2008), KL-UCRL (Filippi et al., 2010), and UCRL3 (Bourel et al., 2020). Since the proposed algorithms are discrete-time rather than continuous-time algorithms, we use a uniformization parameter equal to the maximal event rate. To find the empirical impact of monotonicity enforcement, we also include an ablation of the proposed method that does not truncate the rates. In other words, it uses the model with the highest maximal gain within the confidence set.

To better compare heterogeneous models, we use the normalized regret. Let $M$ be the model, $k(t)$ be the episode step $t$ falls in, and $T$ be the time horizon, this is equal to

$$\text{NormRegret}(M, T) = \sum_{t=1}^{T} \frac{\Delta_{t,k(t)}}{g_M^*}$$

The normalized regret, over time, is presented in Figure 1. The average value of the ratio of the total reward between the selected learning algorithm and the optimal policy is presented in Table 1. We note that regret is approximately 30% higher for the ablation when compared to UCRL-TSAC at 11 states, which increases to approximately 100% at 51 states. There are two reasons for this, one is that the bias in the extended model has tighter bounds and the second is that the truncation provides valuable information about less-seen states when rate monotonicity is known. It is not necessarily possible to disentangle these two effects, but monotonicity enforcement explicitly rules out optimistic policies that rely on distant but high-probability states, which have the largest impact on both the bias and the regret.

It is clear that KL-UCRL achieves sublinear regret at 11 states, and UCRL3 achieves it at up to 21 states. However, in all other cases the general purpose algorithms appear to fail at achieving sublinear regret within the time horizon, unlike the proposed UCRL-TSAC algorithm. There are three explanations for this. One is that the transitions are structured, as the system may only transition from state $s$ to state $s - 1$ or $s + 1$, and information can be shared between different actions. This is ameliorated in part by the use of Kullback-Leibler divergence in the KL-UCRL algorithm and the support estimation of UCRL3. The second is that these algorithms are discrete-time algorithms and require the use of uniformization. A uniformized CTMDP will not have identical transient behavior to the CTMDP itself (Puterman, 2005), although this effect may be small. The third is that the bias in the extended model may be large, as there is no guarantee that the bias of the optimistic model or the true model will be small under the policy in each episode.

## 6 CONCLUSION AND FUTURE WORK

In conclusion, we have presented a reinforcement learning algorithm for admission control in two-sided queues with quite general assumptions on the rates and rewards of the system. We have derived bounds on the expected regret that are independent of the diameter and are polynomial in the number of states. Our results indicate that reinforcement learning algorithms can work well on queueing systems even if rates can be arbitrary and state-dependent. One promising direction for future work include extending these results to admission in queueing networks by applying Norton's theorem (Chandy et al., 1975), similar to the approach in (Anselmi et al., 2024). Another promising direction would be to extend this work to cases in which customer-server compatibilities may exist, using the product-form and quasi-reversibility of order-independent loss queues.

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

## A   Behavior under the Optimal Policy

We consider a more general version of Assumption 1 for this section, as we use it extensively in the Appendix. In particular, the bias bounds also hold if the maximal rate from $s$ to $s + 1$ is decreasing over $s$, and the reverse for the rate from $s$ to $s-1$. This is more general than the previous assumption, since if both $\Lambda(s)$ and $\gamma(s)$ are individually increasing over $s$, then $\Lambda(s) + \gamma(s)$ is increasing as well. The same holds for $M(s)$ and $\eta(s)$. Therefore, the assumption given below is strictly implied by Assumption 1, as presented in the main paper. The extended model, presented in Section B, is guaranteed to fulfill this assumption as opposed to the stronger Assumption 1, which is assumed to hold for the true model.

**Assumption 2.** *The sum of total customer arrival rate and server abandonment rate, $\Lambda(s) + \eta(s)$, is non-increasing over $s$. The sum of total server arrival rate and customer abandonment rate, $M(s) + \gamma(s)$, is non-decreasing over $s$. Separately, $\eta(s)$ is non-increasing over $s$, and $\gamma(s)$ is non-decreasing over $s$.*

The results in this section are presented in terms of a single generic model. For notational simplicity, we do not include subscripts indicating the model used in this section since we do not need to compare different models with each other. Furthermore, the model formulation is symmetric. A particular model can be matched to another one in which customers become servers, and vice versa, by reflecting the state space around $0$ and renaming arrival and abandonment rates. Therefore, we omit arguments that can be established with little effort other than renaming parameters and changing signs. More complex arguments are included, such as the proof of Lemma 4, even if they are identical in essence to a previous one.

We also include the assumption that all rates are bounded explicitly here, as not all models that fall within the confidence set will fulfill this assumption. This assumption is somewhat more general than the formulation given in the paper, in that we assume that $\Lambda(s) + \eta(s) \leq \Lambda^{\max} + \eta^{\max}$, rather than $\lambda(s) \leq \Lambda^{\max}$. The likewise is assumed with respect to $M(s)$ and $\gamma(s)$. This is because the extended model, which is presented later in Section 2 to find an optimistic policy, may violate individual bounds over $\Lambda^{\max}$ and $M^{\max}$, but not the cumulative bounds $\Lambda^{\max} + \eta^{\max}$ and $M^{\max} + \gamma^{\max}$. However, this more general formulation also gives linear bounds for the bias, with regards to the state space.

**Assumption 3.** *There exists a known value $q^{\min}$, such that $\gamma(s) > q^{\min}$ for all $s > 0$ and $\eta(s) > q^{\min}$ for all $s < 0$. Furthermore, for all states $s$, $\Lambda(s) + \eta(s) > q^{\min}$ and $M(s) + \gamma(s) > q^{\min}$. Similarly, for all states $s$, there exist finite upper bounds such that all aggregate total rates are bounded above, $\Lambda(s) + \eta(s) \leq \Lambda^{\max} + 1_{s<0}\eta^{\max}$, $M(s) + \gamma(s) \leq M^{\max} + 1_{s>0}\gamma^{\max}$. Furthermore, the abandonment rates are bounded above. $\eta(s) \leq \eta^{\max}$, and $\gamma(s) \leq \gamma^{\max}$.*

The true model, which is the unknown model that accurately represents the system to be learned, is assumed to fulfill assumptions 1 and 3, and therefore fulfills Assumption 2 as well.

**Proposition 6.** *The MDP is unichain, and $0$ is always a recurrent state under every policy $\pi$.*

*Proof.* In order to show this, it is just necessary to show that a single recurring class exists for every policy. We note that the state space is equal to $\mathbb{Z} \cap [\underline{s}, \bar{s}]$, and is therefore finite. Therefore, there exists at least one positive recurrent class of states. Since the abandonment rates are strictly positive, the transition rate from $s$ to $s - 1$ is strictly positive for all $s > 0$ under every policy, and likewise with the transition rate from $s$ to $s + 1$ where $s < 0$. Therefore, under every policy there exists a single recurring class containing the state $0$. $\qquad\square$

**Proposition 1.** *Under any policy $\pi$, the probability of being in state $s$ in steady-state is equal to*

$$p_\pi^\infty(s) = \begin{cases} \left(1 + \sum_{s''=1}^{\bar{s}} \prod_{s'=1}^{s''} \frac{\lambda_\pi(s'-1)}{\mu_\pi(s')+\gamma(s)} + \sum_{r=0}^{-\underline{s}} \prod_{s'=s''}^{-1} \frac{\mu_\pi(s+1)}{\lambda_\pi(s)+\eta(s)}\right)^{-1} & s = 0 \\ p_\pi^\infty(0) \prod_{s'=1}^{s} \frac{\lambda_\pi(s'-1)}{\mu_\pi(s')+\gamma(s)} & 0 < s \leq \bar{s} \\ p_\pi^\infty(0) \prod_{s'=s}^{-1} \frac{\mu_\pi(s+1)}{\lambda_\pi(s)+\eta(s)} & \underline{s} \leq s < 0 \end{cases}$$

*Proof.* This follows from an elementary induction argument, as is standard in single-class reversible queues, and then renormalizing around the probability at state $0$. Similar results can be found in

(Bhat, 2008; Bolch et al., 2006) for elementary $M/M/1$ queueing systems, with a straightforward generalization to two-sided queues. □

## A.1 BOUNDS ON THE BIAS

We now present several results that establish that under Assumption 2, the bias is linearly bounded over the number of states. Since the regret of the learning algorithm presented later scales along with the bias, this is necessary to get sub-exponential regret over the state space. We begin by presenting a closed-form expression for the relative bias, $\Delta h_\pi(s)$, which is the forward difference of the bias between two states.

**Proposition 2.** *At any state $s$ such that $s$ and $s + 1$ are recurrent, the relative bias*

$$\Delta h_\pi(s) = h_\pi(s + 1) - h_\pi(s)$$

*is equal to*

$$\Delta h_\pi(s) = \frac{1}{p^\infty(s)(\lambda_\pi(s) + \eta(s))} \sum_{s' > s} p_\pi^\infty(s')[\bar{r}_\pi(s') - g_\pi] \tag{5}$$

*Proof.* For equation (5), we proceed by induction from the highest recurrent state, which we label as $s^{\max}$. By the gain bias equations, we have

$$(\mu_\pi(s^{\max}) + \gamma(s^{\max}))\Delta h_\pi(s^{\max} - 1) = \bar{r}_\pi(s^{\max}) - g_\pi$$

Therefore

$$\Delta h_\pi(s^{\max} - 1) = \frac{1}{\mu_\pi(s^{\max}) + \gamma(s^{\max})}(\bar{r}_\pi(s^{\max}) - g_\pi)$$

$$= \frac{1}{p_\pi^\infty(s^{\max} - 1)(\lambda_\pi(s^{\max} - 1) + \eta(s^{\max} - 1))}p_\pi^\infty(s^{\max})(\bar{r}_\pi(s^{\max}) - g_\pi)$$

For the induction step, let (5) be true for all $s' > s$. By the gain-bias equations

$$(\lambda_\pi(s + 1) + \eta(s + 1))\Delta h_\pi(s + 1) - (\mu_\pi(s + 1) + \gamma(s + 1)\Delta h_\pi(s) = g_\pi - \bar{r}_\pi(s + 1)$$

This implies

$$(\mu_\pi(s + 1) + \gamma(s + 1))\Delta h_\pi(s) = \bar{r}_\pi(s + 1) - g_\pi - \frac{1}{p^\infty(s + 1)} \sum_{s' > s+1} p_\pi^\infty(s')[\bar{r}_\pi(s') - g_\pi]$$

After simplifying, we derive

$$\Delta h_\pi(s) = \frac{1}{\mu_\pi(s + 1) + \gamma(s + 1)}(\bar{r}_\pi(s + 1) - g_\pi)$$

$$- \frac{1}{(\mu_\pi(s + 1) + \gamma(s + 1))p^\infty(s + 1)} \sum_{s' > s+1} p_\pi^\infty(s')[\bar{r}_\pi(s') - g_\pi]$$

Finally, noting that the induced CTMC is time-reversible over the set of recurrent states, we have $(\mu_\pi(s + 1) + \gamma(s + 1))p_\pi^\infty(s + 1) = (\lambda_\pi(s) + \eta(s))p_\pi^\infty(s)$, and therefore

$$\Delta h_\pi(s) = \frac{1}{p^\infty(s)(\lambda_\pi(s) + \eta(s))} \sum_{s' > s} p_\pi^\infty(s')[\bar{r}_\pi(s') - g_\pi]$$

□

The next lemma is quite straightforward. In essence, this proof establishes a direct relationship between the optimal policy and the relative bias.

**Lemma 1.** *Let Assumption 2 hold. Consider a deterministic gain-optimal policy $\pi$, and a recurrent state $s$ such that $s + 1$ is also recurrent. Let $a(s)$ and $a(s + 1)$ be the corresponding actions where $\pi(s, a) = 1$ and $\pi(s + 1, a) = 1$. For all customer types $i$ such that $i \notin a_1$*

$$\Delta h_\pi(s) + r_i^c(s) \leq 0 \tag{6}$$

*For all customer types $i$ such that $i \in a_1$*

$$\Delta h_\pi(s) + r_i^c(s) \geq 0 \tag{7}$$

*Similarly, for all server types $j$ such that $j \notin a_2$*

$$\Delta h_\pi(s) - r_j^s(s+1) \geq 0 \tag{8}$$

*Finally, for all server types $j$ such that $j \in a_2$*

$$\Delta h_\pi(s) - r_j^s(s+1) \leq 0 \tag{9}$$

*Proof.* Let $\pi^*$ be a gain-optimal deterministic policy. Let $\mathbf{Z}_\pi$ be the generator matrix for the induced CTMC under policy $\pi$. Then, for any recurrent state $s'' \in \mathcal{S}_{\pi^*}^+$, $h_\pi(s') = h_{\pi^*}(s')$, since $Z_{\pi^*,s',s''} = Z_{\pi,s',s''} = 0$ if $s''$ is not recurrent.

Consider the following expression for the bias $h_\pi(s)$ and average reward $\bar{r}(s,a)$

$$h_\pi(s) = \frac{1}{\lambda_\pi(s) + \eta(s) + \mu_\pi(s) + \gamma(s)} \Bigg[ \lambda_\pi(s)h(s+1) + \eta(s)h(s+1)$$

$$+ \mu_\pi(s)h(s-1) + \gamma(s)h(s-1) + \bar{r}(s,a) - g_\pi \Bigg]$$

$$\bar{r}(s,a) = \lambda_\pi(s)r^c(s) + \mu_\pi(s)r^s(s) + r^a(s)\eta(s) + r^a(s)\gamma(s)$$

Consider a first-degree rational function.

$$f(t) = \frac{at + b}{t + c}$$

This is increasing on $t \geq 0$ if $a \geq f(0)$, and decreasing if $a \leq f(0)$.

To prove (6) and (7), we can consider two policies $\pi$ and $\pi'$, that differ only in that $\pi$ accepts a customer type $i$ in state $s$ and $\pi'$ does not. Let $a$ and $a'$ be the corresponding actions in state $s$. Then, we define the coefficients for a first degree rational function below

$$a = h(s+1) + r_i^c$$

$$b = \left( \lambda_{\pi'}(s) + \mu_{\pi'}(s) + \eta(s) + \gamma(s) \right) h_{\pi'}(s)$$

$$c = \lambda_{\pi'}(s) + \mu_{\pi'}(s) + \eta(s) + \gamma(s)$$

Noting that $h_\pi(s) = f(\lambda_i)$ and $h_{\pi'}(s) = f(0)$, (6) and (7) immediately follow. A substantially similar argument, using $a = h(s-1) + r_j^s$ and evaluating at $f(\mu_j)$, establishes (9) and (8) as well. $\quad\square$

The following corollary enables a simplification of the action space to the reduced action space $\tilde{\mathcal{A}}$. Since there exists an optimal policy that requires all types to exceed a given threshold, we only need to consider actions for which either no customers are accepted, or there exists a threshold customer type $i$ for each state, for which all types with a greater or equal reward are accepted and none with a lower reward are. Likewise, either no servers should be accepted in a given state, or there similarly exists a threshold server type $j$.

**Corollary 1.** *There exists a deterministic gain-optimal policy in which for each state $s$, there are threshold customer types $T^c(s)$ and service types $T^s(s)$ such that all customer types with a reward greater than or equal to $r_{T^c(s)}^c(s)$ are accepted, and all services types with a reward greater than or equal to $r_{T^s(s)}^s(s)$ are accepted. No types with a lower reward are accepted. Furthermore, no customers are accepted in positive transient states and no servers are accepted in negative transient states.*

*Proof.* This follows immediately from noting a deterministic gain-optimal policy exists, and from Lemma 1. This is shown by finding the customer type with the lowest reward that is greater than

$-\Delta h_\pi(s)$, and the server type with the lowest reward that is greater than $\Delta h_\pi(s-1)$. An otherwise-identical policy with the same recurrent class that does not accept customers (servers) in positive (negative) transient states has identical gain, and is therefore also gain-optimal. Since the MDP is unichain, the recurrent set cannot change without a change in the actions within the recurrent set. Therefore, a policy with these properties must exist. $\qquad\square$

The next lemma establishes that given arbitrary acceptance rates that are less than the arrival rate in each state, a corresponding policy can be constructed. Although straightforward, this is useful for proving optimism, by establishing the existence of policies that have equal rates but a better reward distribution.

**Lemma 2.** *Consider a set of target rates for each state $s$ and customer type $i$, where $0 \le \lambda_i^{tgt}(s) \le \lambda_i(s)$. Likewise for each state $s$ and server type $j$ there are target rates $0 \le \mu_j^{tgt}(s) \le \mu_j(s)$ for each state, then there exists a policy $\pi$ such that for all states $s$, customer types $i$ and server types $j$*

$$\lambda_{i,\pi}(s) = \sum_{i\in a_1} \pi(s,a)\lambda_i(s) = \lambda_i^{tgt}(s)$$

$$\mu_{j,\pi}(s) = \sum_{j\in a_2} \pi(s,a)\mu_j(s) = \mu_j^{tgt}(s) \tag{10}$$

*Proof.* We prove this for an arbitrary state $s$. We use a recursive argument over the number of customer and server types with $\lambda_i^{tgt}(s) > 0$ and $\mu_j^{tgt}(s) > 0$, respectively. We begin by noting that the number of customer and server types are finite, that there exists a policy $\pi_0$ where $\lambda_{i,\pi_0}(s) = 0$ and $\mu_{j,\pi_0}(s) = 0$ for all customer types $i$, server types $j$ and states $s$.

Then, assume that there exists a policy $\pi'$ that fulfills (10), where there exists some $i$ such that for all states $s$ and $l \ne i$

$$\lambda_{l,\pi'}(s) = \lambda_l^{tgt}(s)$$

$$\lambda_{i,\pi'}(s) = 0$$

Then, we can construct a new policy $\pi$ as follows. For all states $s$

$$\pi(s,a) = \left(1 - \frac{\lambda_i^{tgt}(s)}{\lambda_i(s)}\right)\pi'(s,a) \quad i \notin a_1$$

$$\pi(s,a) = \frac{\lambda_i^{tgt}(s)}{\lambda_i(s)}\pi'(s,a\backslash\{i\}) \quad i \in a_1$$

Then,

$$\lambda_{l,\pi}(s) = \lambda_l(s) \sum_{a|l\in a_1} \pi(s,a) = \lambda_l(s) \sum_{a|l\in a_1, i\notin a_1} \left(1 - \frac{\lambda_i^{tgt}(s)}{\lambda_i(s)} + \frac{\lambda_i^{tgt}(s)}{\lambda_i(s)}\right)\pi'(s,a)$$

$$= \lambda_{l,\pi'}(s) = \lambda_l^{tgt}(s)$$

$$\lambda_{i,\pi}(s) = \lambda_i(s) \sum_{a|i\in a_1} \pi(s,a) = \lambda_i(s) \sum_{a|i\in a_1} \frac{\lambda_i^{tgt}}{\lambda_i(s)}\pi'(s,a\backslash\{i\}) = \lambda_i^{tgt}(s)$$

$$\mu_{l,\pi}(s) = \mu_l(s) \sum_{a|l\in a_2} \pi(s,a) = \mu_l(s) \sum_{a|l\in a_2, i\notin a_1} \left(1 - \frac{\lambda_i^{tgt}(s)}{\lambda_i(s)} + \frac{\lambda_i^{tgt}(s)}{\lambda_i(s)}\right)\pi'(s,a)$$

$$= \mu_{l,\pi'}(s) = \mu_l^{tgt}(s)$$

The same reasoning establishes this property when there exists exactly one server type $j$ where $\mu_{j,\pi'}(s) = 0$ for all $s$ instead. This completes the proof. $\qquad\square$

The following results establish that the relative bias, $\Delta h(s)$, has a linear bound over the size of the state space. Since this is a multiplier on the dominating term of the regret bound, it implies

that the algorithm presented is capable of learning at a significantly greater number of states than general-purpose reinforcement learning algorithms, as for the latter ones the diameter is exponential in general with regards to the state space (Weber et al., 2024).

**Lemma 3.** *Let Assumption 2 hold. Assume that all rates are bounded below by* 1. *Then, for any gain-optimal deterministic policy* $\pi$ *and any state* $s$ *such that both* $s$ *and* $s + 1$ *are recurrent. Let* $r^{\max} \geq q^{\max}$ *be an upper bound for the mean reward in each state. Then, the following holds for every recurrent state* $s$

$$\Delta h_\pi(s) \leq 2Sr^{\max}$$

*Proof.* Let $a(s)$ and $a(s + 1)$ be the actions such that $\pi(s, a(s)) = 1$ and $\pi(s + 1, a(s + 1)) = 1$. For convenience, we use

$$\rho(s) = \frac{\lambda_\pi(s - 1) + \eta(s - 1)}{\mu_\pi(s) + \gamma(s)}$$

Let $I$ be the set of all states $s$ such that both $s$ and $s + 1$ is recurrent. Since the MDP is unichain, and the state space is a subset of $\mathbb{Z}$, $I$ is an interval. Next, we split $I$ into three sets, $A$, $B$, $C$. Let $A$ be all states $s \in I$ such that there exists some arrival type $i$ such that $i \notin a(s)_1$, or there is a departure type $j$ such that $j \in a(s + 1)_2$. Let $B$ include all states $s \in D \backslash A$ such that $\rho(s) \geq 1$, and $C$ include all states $s \in D \backslash A$ such that $\rho(s) < 1$. For all $s \in B \cup C$, it should be noted that $\lambda_\pi(s) = \Lambda(s)$ and $\mu_\pi(s + 1) = 0$. Therefore, for all $s \in B \cup C$

$$\rho(s) = \frac{\Lambda(s) + \eta(s)}{\gamma(s + 1)}$$

Two properties can immediately be established. The first is that for all $s \in B \cup C$

$$a(s)_1 = \{1, \ldots, N^c\}$$
$$a(s + 1)_2 = \emptyset$$

The second is that by Assumption 2, $\max B < \min C$. In other words, all states in $B$ proceed those in $C$.

Now, for all $s \in A$ either there exists some arrival level $i \notin a(s)_1$, in which case

$$\Delta h_\pi(s) \leq -r_i^c(s) \leq 1$$

Otherwise, there exists some service level $j \in a(s + 1)_2$

$$\Delta h_\pi(s) \leq r_j^s(s + 1) \leq 1$$

Therefore, $\Delta h_\pi(s) \leq 1$ for all $s \in A$.

Next, we derive upper bounds on the relative bias for all $s \in A \cup B$. We aim to establish that for all $s \in A \cup B$,

$$\Delta h_\pi(s) \leq 2(s - s_0 + 1)r^{\max}$$

Since we know that this bound holds for all $s \in A$, since $\Delta h_\pi(s) \leq 1 \leq r^{\max}$. If $B = \emptyset$, then the induction hypothesis holds for $A \cup B = A$, since the bound is weaker than the one we already found for $A$. Otherwise, we consider the case that $B$ is non-empty. We then proceed by induction over $s$, beginning with the lowest recurrent state $s_0$, which is known to be in $A \cup B$ when $B \neq \emptyset$, and ending with the maximal value of $B$. If $s_0 \in B$, then using the gain-bias equations we have

$$(\lambda_\pi(s_0) + \eta(s_0))\Delta h(s_0) = g_\pi - \bar{r}_\pi(s_0)$$

$$\Delta h(s_0) = \frac{1}{\lambda_\pi + \eta(s_0)}(g_\pi - \bar{r}_\pi(s_0)) \leq 2r^{\max}$$

Otherwise, $s_0 \in A$, and the proposed bound holds.

For the induction step, if $s \in A$, the proposed bound holds. If $s - 1 \in A$ and $s \in B$, then noting that $\lambda_\pi(s) + \eta(s) \geq 1$ and $g_\pi - \bar{r}_{pi}(s) \leq 2r^{\max}$, we can derive the following from the gain-bias equations

$$(\lambda_\pi(s) + \eta(s))\Delta h(s) - (\mu_\pi(s) + \gamma(s))\Delta h(s - 1) = g_\pi - \bar{r}_\pi(s)$$

$$\Delta h(s) = \frac{\mu_\pi(s) + \gamma(s)}{\lambda_\pi(s) + \eta(s)} \Delta h_\pi(s-1) + \frac{1}{\lambda_\pi(s) + \eta(s)} [g_\pi - \bar{r}_\pi(s)]$$

$$\leq \mu_\pi(s) + \gamma(s) + 2r^{\max}$$

*(since $\Delta h_\pi(s-1) \leq 1$, as $s-1 \in A$, and that $\lambda_\pi(s) + \eta(s) \geq 1$)*

$$\leq (s - s_0 + 1)2r^{\max}$$

*(since $\mu_\pi + \gamma(s) \leq q^{\max} \leq r^{\max}$)*

Otherwise, $s - 1 \in B$, in which case

$$(\lambda_\pi(s) + \eta(s))\Delta h(s) - (\mu_\pi(s) + \gamma(s))\Delta h(s-1) = g_\pi - \bar{r}_\pi(s)$$

$$\Delta h(s) = \frac{\mu_\pi(s) + \gamma(s)}{\lambda_\pi(s) + \eta(s)} \Delta h(s-1) + \frac{1}{\lambda_\pi(s) + \eta(s)} [g_\pi - \bar{r}_\pi(s)]$$

$$\leq \frac{2}{\rho(s)}(s - s_0)r^{\max} + \frac{1}{\lambda_\pi(s) + \eta(s)} [g_\pi - \bar{r}_\pi(s)]$$

*(by Assumption 2 and substituting $\Delta h(s-1)$)*

$$\leq 2(s - s_0)r^{\max} + \frac{1}{\lambda_\pi(s) + \eta(s)} [g_\pi - \bar{r}_\pi(s)]$$

*(By noting that $\rho(s) \geq 1$, since $s \in B$)*

$$\leq 2(s - s_0)r^{\max} + 2r^{\max}$$

$$\leq 2(s - s_0 + 1)r^{\max}$$

Finally, we consider all $s \in C$. Since $\rho(s) < 1$ for all $s \in C$, we have $p_\pi^\infty(s') < p_\pi^\infty(s)$ for all $s' > s$. Then, using (5) in Proposition 2

$$\Delta h_\pi(s) = \frac{1}{p^\infty(s)(\lambda_\pi(s) + \eta(s))} \sum_{s' > s} p_\pi^\infty(s')[\bar{r}_\pi(s') - g_\pi]$$

$$\leq \sum_{s'=s+1}^{k} [\bar{r}_\pi - g_\pi]$$

$$\leq 2Sr^{\max}$$

$\square$

**Lemma 4.** *Let Assumption 2 hold. Assume that $q^{\min} \geq 1$. Then, for any gain-optimal deterministic policy $\pi$ and any state $s$ such that both $s$ and $s + 1$ are recurrent. Let $r^{\max} \geq q^{\max}$ be an upper bound for the mean reward in each state. The following is true for every state $s$.*

$$\Delta h(s) \geq -2Sr^{\max}$$

*Proof.* The proof of this is similar, in essence, to the one given above for the upper bound. We partition $D$ into $A'$, $B'$, and $C'$. $A'$ contains all states $s \in D$ such that there exists some arrival type $i$ such that $i \in a(s)_1$, or there exists some arrival type $j$ such that $j \in a(s+1)_2$. $B'$ is equal to all sets $s \in D \backslash A'$ such that $\rho(s) \leq 1$, and $C'$ is the remaining sets in $s \in D \backslash A'$ such that $\rho(s) > 1$. Note that in non-trivial cases $B$ will be disjoint from $B'$, and likewise with $C$ and $C'$. However, the role of each set remains the same in the proof. Likewise for $s \in A'$ either there exists some $i \in a(s)_1$, or some service level $j \notin a(s+1)_2$. Therefore one of the following inequalities holds, both of which result in the bound $\Delta h(s) \geq -1$

$$\Delta h(s) \geq -r_i^c(s) \geq -1$$

$$\Delta h(s) \geq r^s(s+1) \geq -1$$

Next, we consider all $s \in B'$. If $B'$ is empty, this is vacuously true. Otherwise, we consider the case that it is non-empty. Similarly, it can be established that the $\min B' > \max C'$. Therefore, we

begin from the highest value in $D$, which we label as $s^{\max}$. Our induction hypothesis is that for all $s \in A' \cup B'$

$$\Delta h(s) \geq -2(s^{\max} - s + 1)r^{\max} \tag{11}$$

If $s^{\max} \in B'$, then

$$(\mu_\pi(s^{\max} + 1) + \gamma(s^{\max} + 1))\Delta h(s^{\max}) = \bar{r}_\pi(s^{\max} + 1) - g_\pi$$

Therefore,

$$\Delta(s^{\max}) = \frac{\bar{r}_\pi(s^{\max} + 1) - g_\pi}{\mu_\pi(s^{\max} + 1) + \gamma(s^{\max} + 1)}$$
$$\geq -2r^{\max}$$

Otherwise, $s^{\max} \in A'$ and the bound holds as well. For the induction step, assume that (11) holds for $s + 1$. Then, if $s + 1 \in A'$

$$(\mu_\pi(s + 1) + \gamma(s + 1))\Delta h(s) - (\lambda_\pi(s + 1) + \eta(s + 1))\Delta h(s + 1) = \bar{r}_\pi(s + 1) - g_\pi$$

$$\Delta h(s) = \frac{1}{\mu_\pi(s + 1) + \gamma(s + 1)}[(\lambda_\pi(s + 1) + \eta(s + 1))\Delta h(s + 1) + \bar{r}_\pi(s + 1) - g_\pi]$$

$$\geq \frac{1}{\mu_\pi(s + 1) + \gamma(s + 1)}[-(\lambda_\pi(s + 1) + \eta(s + 1)) - 2r^{\max}]$$

*(Since $\Delta h_\pi(s) \leq 1$, which is implied directly by Lemma 1 and that $s + 1 \in A'$)*
$$\geq -(\lambda_\pi(s + 1) + \eta(s + 1)) - 2r^{\max}$$
$$\geq -3r^{\max}$$
$$\geq -2(s^{\max} - s + 1)r^{\max}$$

Otherwise, if $s + 1 \in B'$

$$\Delta h(s) = \frac{(\lambda_\pi(s + 1) + \eta(s + 1))\Delta h(s + 1)}{\mu_\pi(s + 1) + \gamma(s + 1)} + \frac{\bar{r}_\pi(s + 1) - g_\pi}{\mu_\pi(s + 1) + \gamma(s + 1)}$$

$$\geq \frac{(\lambda_\pi(s + 1) + \eta(s + 1))}{\mu_\pi(s + 1) + \gamma(s + 1)}(-2(s^{\max} - (s + 1) + 1)r^{\max})$$

$$+ \frac{\bar{r}_\pi(s + 1) - g_\pi}{\mu_\pi(s + 1) + \gamma(s + 1)}$$

*(Substituting $\Delta h(s + 1)$ according to (11))*

$$\geq \frac{(\lambda_\pi(s) + \eta(s))}{\mu_\pi(s + 1) + \gamma(s + 1)}(-2(s^{\max} - (s + 1) + 1)r^{\max})$$

$$+ \frac{\bar{r}_\pi(s + 1) - g_\pi}{\mu_\pi(s + 1) + \gamma(s + 1)}$$

*(Applying Assumption 2 and noting that it is multiplied by a negative quantity)*

$$\geq \frac{(\lambda_\pi(s) + \eta(s))}{\mu_\pi(s + 1) + \gamma(s + 1)}(-2(s^{\max} - (s + 1) + 1)r^{\max})$$

$$- 2r^{\max})$$

*(Noting that $\mu_\pi(s + 1) + \gamma(s + 1) \geq 1$)*
$$\geq (-2(s^{\max} - (s + 1) + 1)r^{\max}) - 2r^{\max})$$

*(Since $s + 1 \in B'$, it follows that $\rho(s + 1) \leq 1$. Note that the coefficient is negative)*
$$\geq (-2(s^{\max} - s + 1)r^{\max})$$

This concludes the proof for $A'$ and $B'$. For $C'$, we know that

$$\Delta h_\pi(s) = \frac{1}{p^\infty(s)(\lambda_\pi(s) + \eta(s))} \sum_{s' > s} p_\pi^\infty(s')[\bar{r}_\pi(s') - g_\pi]$$

We can use the identity $\sum_{s'} p_\pi^{\infty(s')} \bar{r}_\pi(s') = g_\pi$ to reverse this.

$$\Delta h_\pi(s) = \frac{1}{p^\infty(s)(\lambda_\pi(s) + \eta(s))} \sum_{s' \leq s} p_\pi^\infty(s')[g_\pi - \bar{r}_\pi(s')]$$

$$\geq \frac{1}{p^\infty(s)(\lambda_\pi(s) + \eta(s))} \sum_{s' \leq s} p_\pi^\infty(s')[-2r^{\max}]$$

$$\geq \frac{1}{p^\infty(s)} \sum_{s' \leq s} p_\pi^\infty(s')[-2r^{\max}]$$

$$\geq -2Sr^{\max}$$

This completes the proof. $\square$

Next, we restate and complete the proof of Proposition 3, from Section 4 of the paper.

**Proposition 3.** *Let Assumption 2 hold. Then, consider a gain-optimal policy $\pi^*$ in which no customer arrivals are accepted in positive transient states, and no server arrivals are accepted in negative ones. Then, for any state $\underline{s} \leq s < \bar{s}$*

$$|\Delta h_{\pi^*}(s)| \leq \frac{2(q^{\max} + 1)}{q^{\min}} S$$

*Proof.* First, we begin by assuming that all rates are bounded below by 1, and all rewards are bounded within the interval $[-1, 1]$, as was done in Lemmas 3 and 4. In particular, we show that in this case

$$|\Delta h_{\pi^*}(s)| \leq 2Sr^{\max}$$

Lemmas 3 and 4 handle the case in which both $s$ and $s + 1$ are both recurrent. Next, we prove this inequality holds for adjacent pairs of states where at least one is transient. Beginning with positive states $s$ that are transient, we have

$$(\mu_{\pi^*}(s) + \gamma(s))\Delta h(s - 1) = \bar{r}_{\pi^*}(s) - g_{\pi^*}$$

Therefore

$$|\Delta h(s - 1)| = \frac{1}{\mu_\pi(s) + \gamma(s)} |\bar{r}_{\pi^*}(s) - g_{pi^*}|$$

$$\leq \frac{2}{\mu_{\pi^*}(s) + \gamma(s)} r^{\max}$$

$$\leq 2Sr^{\max}$$

Similarly, we can apply the same argument to cases in which $s < 0$ and $s$ is transient.

$$(\lambda_{\pi^*}(s) + \rho(s))\Delta h(s) = g_{\pi^*} - \bar{r}_{\pi^*}(s)$$

$$|\Delta h(s - 1)| = \frac{1}{\lambda_{\pi^*}(s) + \rho(s)} |\bar{r}_\pi(s) - g_\pi|$$

$$\leq \frac{2}{\lambda_{\pi^*}(s) + \rho(s)} r^{\max}$$

$$\leq 2Sr^{\max}$$

Then, applying the upper bound $r^{\max} = q^{\max} + 1$, which applies since all reward parameters are bounded within $[-1, 1]$, we complete the proof for the case where all rates are greater than 1. Next, we generalize this result to cases in which there may be some rates that are less than 1 by scaling the times and rewards.

We consider a time-scaling factor of $1/q^{\min}$. Note that in order to keep the reward vector constant, we also need to scale the instantaneous rates $r^s(s), r^c(s), r^a(s)$ by $q^{\min}$. This does not interfere with

the assumption that all rates remain within $[-1, 1]$, since we have already assumed that $q^{\min} < 1$. Note that the holding reward, $r^h(s)$, is semantically different since it is a rate reward rather than a transition reward, and it can be left unchanged. Therefore, using $\bar{r}'(s, a)$ to denote the time and reward-scaled average reward in each state

$$\bar{r}'(s, a) = c^h(s) + \frac{1}{q^{\min}} \gamma(s)(q^{\min} r^a(s)) + \frac{1}{q^{\min}} \eta(s)(q^{\min} r^a(s))$$

$$+ \sum_{i \in a_1} \frac{1}{q^{\min}} \lambda_i(s)(q^{\min} r_i^c) + \sum_{j \in a_2} \frac{1}{q^{\min}} \mu_i(s)(q^{\min} r_i^s)$$

$$= \bar{r}(s, a)$$

The new induced generator matrix is $\boldsymbol{Z}'_\pi = \frac{1}{q^{\min}} \boldsymbol{Z}_\pi$. Let $\boldsymbol{h}'_\pi$ be the bias under the time and reward-scaling regime we've described earlier. Since the condition that all rates must be greater than 1 now applies

$$|\Delta h'_{pi^*}(s)| \leq 2S(q^{\max} + 1)$$

Up to a constant, $\boldsymbol{h}'_{\pi^*}$ is the unique solution of the following equation

$$\frac{1}{q^{\min}} \boldsymbol{Z}_{\pi^*} \boldsymbol{h}'_{\pi^*} = \mathbf{1} g_{\pi^*}(s) - \boldsymbol{r}_{\pi^*}(s)$$

Therefore

$$\boldsymbol{h}_{\pi^*} = \frac{1}{q^{\min}} \boldsymbol{h}'_{\pi^*}$$

This immediately implies that

$$\Delta h_{\pi^*}(s) \leq \frac{2}{q^{\min}} S(q^{\max} + 1)$$

Then, by a different time-scaling argument we can rescale when $q^{\min} > 1$ to achieve tighter bounds. We again use a time-scaling factor of $\frac{1}{q^{\min}}$, but do not rescale the instantaneous rewards. However, $r^h$ should be scaled by $\frac{1}{q^{\min}}$, which again preserves the bounds within $[-1, 1]$ since $\frac{1}{q^{\min}} < 1$. Then it can be established that under this scaling regime, $\bar{r}'(s, a) = \frac{1}{q^{\min}} r(s, a)$. By the linearity of the gain-bias equations the bias under the scaled system is identical to that of the original system. However, in the scaled system we note that $\frac{q^{\max} + 1}{q^{\min}}$ is an appropriate upper bound for the reward, since the mean reward vector and rates are scaled. Therefore, we can derive a tighter bound

$$\Delta h_{\pi^*}(s) \leq \frac{2}{q^{\min}} S(q^{\max} + 1)$$

$\square$

The bounds given above do not hold in general, without assumptions on the model. To demonstrate the necessity of Assumption 2, we present a result that demonstrates that the bias bounds do not hold in more general settings. Therefore, we establish that further assumptions on the model are necessary to achieve polynomial bounds on the bias.

**Proposition 4.** *There exist no polynomial bounds for the bias under an optimal policy, with regards to the state space, for models in which only Assumption 3 necessarily holds.*

*Proof.* We proceed by directly constructing a model in which the bias bounds fail. We only use a one-sided queue in this case for simplicity. We assume that this is on the customer side, but the same argument may be used to establish a bound when $\bar{s} = 0$ and the buffer only admits entities on the server side.

Let $\bar{s}$ be an odd upper bound for the state space. Then, let all rewards be equal to 0 except for $r^h(\bar{s}) = 1$. Then, since $\Lambda^{\max} \geq \Lambda(s) > q^{\min}$, because $\eta(s) = 0$ for all positive states, we can

construct the following model using a single customer type 1.

$$\lambda_1(s) = \begin{cases} q^{\min} & s < \lfloor \frac{1}{2}\bar{s} \rfloor \\ \min(\Lambda^{\max}, \gamma^{\max}) & s \geq \lfloor \frac{1}{2}\bar{s} \rfloor \end{cases}$$

$$\gamma(s) = \begin{cases} \min(\Lambda^{\max}, \gamma^{\max}) & s \leq \lfloor \frac{1}{2}\bar{s} \rfloor \\ q^{\min} & s > \lfloor \frac{1}{2}\bar{s} \rfloor \end{cases}$$

It then follows that the only optimal policy $p_{\pi^*}^\infty(s)$ accepts all customer arrivals in every state. Then, using Proposition 1, we have

$$\sum_{s=\lceil \frac{1}{2}\bar{s} \rceil}^{\bar{s}} p_{\pi^*}^\infty(s) = \sum_{s=0}^{\lfloor \frac{1}{2}\bar{s} \rfloor} p_{\pi^*}^\infty(s)$$

Then at $\lfloor \frac{1}{2}\bar{s} \rfloor$

$$\left| \Delta h_{\pi^*}(\lfloor \frac{1}{2}\bar{s} \rfloor) \right| = \left| \frac{1}{p_{\pi^*}^\infty(\lfloor \frac{1}{2}\bar{s} \rfloor)} \sum_{s=\lceil \frac{1}{2}\bar{s} \rceil}^{\bar{s}} p_{\pi^*}^\infty(s)(\bar{r}_{\pi^*}(s) - g_{\pi^*}) \right|$$

$$\leq \left| \frac{1}{p_{\pi^*}^\infty(\lfloor \frac{1}{2}\bar{s} \rfloor)} (g_{\pi^*} - \frac{1}{2}g_{\pi^*}) \right|$$

*(Noting that the only state with non-zero reward is state $\bar{s}$,*

*and exactly half of the probability density occurs at states above $\lfloor \frac{1}{2}\bar{s} \rfloor$)*

$$\leq \frac{p_{\pi^*}^\infty(\bar{s})}{2p_{\pi^*}^\infty(\lfloor \frac{1}{2}\bar{s} \rfloor)}$$

*(Noting that $g_{\pi^*} = p_{\pi^*}^\infty(s)$)*

$$= \frac{1}{2} \left( \frac{\min(\Lambda^{\max}, \gamma^{\max})}{q^{\min}} \right)^{\lfloor \frac{1}{2}\bar{s} \rfloor}$$

*(Applying Proposition 1)*

Since $\frac{\min(\Lambda^{\max}, \gamma^{\max})}{q^{\min}} > 1$, we have an exponential bound on the relative bias, and therefore the bias, with respect to the state space. This completes the proof. □

# B  FINDING AN OPTIMISTIC POLICY

## B.1  CONFIDENCE INTERVALS

In this section, we first restate the confidence intervals for the inter-event rate of positive and negative events, as well as the conditional event probabilities. We then use these to construct somewhat larger sets for the total arrival rates $\Lambda$ and $M$ individually, as well as for the abandonment rates. For notation, $v_k$ represents the total number of steps within episode $k$, $V_k$ gives the total number of steps up to and including episode $k$. $V_k(s)$ gives the number of steps up to the end of episode $k$ in state $s$, while $v_k(s)$ gives the number of steps within episode $k$ in state $s$. $\tau_t$ is used for the observed sojourn time at step $t$, and $s(t)$ gives the state observed at step $t$. $\delta$ is the global confidence parameter. We use $\mathcal{D}_k$ to represent the confidence set at episode $k$, which is the set of all models that have each parameter within the respective confidence interval.

Analogously to how states are positive when customers are present and negative when servers are, we use a similar convention to indexing event types. Each arrival $l \in \{1, \dots, N^c\}$ corresponds to a customer arrival of type $l$, and $l \in \{-N^s, \dots, -1\}$ corresponds to a server arrival of type $-l$. For notational convenience, we use $l = 0$ to represent an abandonment.

We then categorize each event based on the potential state change it can induce. Let $B^+(s)$ be the set of all event types that potentially lead, depending on the action taken, from $s$ to $s + 1$, namely all customer arrivals as well as abandonments when $s < 0$. This is equal to $\{l | l \geq 0\}$ if $s < 0$ and

$\{l|l > 0\}$ if $s \geq 0$. Likewise, let $B^-(s)$ be the set of all event types that potentially lead from $s$ to $s-1$, which is equal to $\{l|l \leq 0\}$ if $s > 0$ and $\{l|l < 0\}$ if $s \leq 0$.

In order to analyze the a particular trajectory of the system, we define a few more quantities. $V_k^+(s)$ be the number of event types in $B^+(s)$ (not necessarily accepted) observed in state $s$ in and before episode $k$. Likewise, we define $V_{k-1}^-(s)$ as the number of event types in $B^-(s)$ observed in state $s$ in and before episode k. For the truncated empirical mean, we also define $V_{k-1,t}^+(s)$ and $V_{k-1}^-(s)$ as the number of event types in $B^+(s)$ and $B^-(s)$, respectively, observed in state $s$ before episode $k$ and step $t$. We also define the following sets

$$T^+(s) = \{t|s(t) = s, l(t) \in B^+(s(t))\}$$
$$T^-(s) = \{t|s(t) = s, l(t) \in B^-(s(t))\}$$

Finally, for each state $s$ let $\tau_t^+(s)$ and $\tau_t^-(s)$ be the observed inter-arrival times of events in $B^+(s)$ and $B^-(s)$, respectively, at $t$.

$$\tau_t^+ = \begin{cases} \sum_{i=1}^t 1_{s(i)=s(t)}\tau_t & t = \{T^+(s(t))\}_0 \\ \sum_{i=1}^t 1_{s(i)=s(t)}\tau_t - \tau_{pred_{T^+(s(t))}}^+(s) & \text{otherwise} \end{cases}$$

$$\tau_t^- = \begin{cases} \sum_{i=1}^t 1_{s(i)=s(t)}\tau_t & t = \{T^-(s(t))\}_0 \\ \sum_{i=1}^t 1_{s(i)=s(t)}\tau(s) - \tau_{pred_{T^-(s(t))}}^+ & \text{otherwise} \end{cases}$$

It can be established, with a simple algebraic argument, that $\tau_t^+$ is an i.i.d exponentially distributed random variable with rate $\Lambda(s) + \eta(s)$ for all $t \in T^+(s)$, and likewise for $\tau_t^-$ with rate $M(s) + \gamma(s)$ for all $t \in T^-(s)$. The next task is to establish the times between events within both $B^+(s)$ and $B^-(s)$. Let $l(t)$ be the event at step $t$. We use the truncated empirical mean, as presented in Lemma 1 of (Bubeck et al., 2013), to estimate the total event rate in each state. This estimator has been used before in continuous-time reinforcement learning (Weber et al., 2024; Gao & Zhou, 2024). Let $\delta_k = \frac{\delta}{V_{k-1}}$ be the confidence parameter used in episode $k$. We present the following estimators for the time between events in $B^+(s)$ and those in $B^-(s)$. Then, the time between events in $B^+(s)$, as well as within $B^-(s)$, can be estimated using the truncated empirical mean. We represent both quantities by $\hat{\tau}_k^+(s)$ and $\hat{\tau}_k^-(s)$, respectively.

$$\hat{\tau}_k^+(s) = \frac{1}{V_{k-1}^+(s)} \sum_{t=1}^{V_{k-1}} \tau_t^+ 1\left\{s(t) = s, l(t) \in B^+(s), \tau_t^+ \leq \sqrt{\frac{2V_{k-1,t}^+(s)}{(q^{\min})^2 \log(\frac{2S}{\delta_k})}}\right\}$$

$$\hat{\tau}_k^-(s) = \frac{1}{V_{k-1}^-(s)} \sum_{t=1}^{V_{k-1}} \tau_t^- 1\left\{s(t) = s, l(t) \in B^-(s), \tau_t^- \leq \sqrt{\frac{2V_{k-1,t}^-(s)}{(q^{\min})^2 \log(\frac{2S}{\delta_k})}}\right\} \quad (12)$$

The confidence half-lengths are given below (Bubeck et al., 2013; Gao & Zhou, 2024). Note that we only need to consider the total number of states, $S$, in the union bound instead of the number of state-action pairs, as all arrivals are observable even if rejected. The following bounds hold with a probability of at least $1 - \frac{\delta}{S}$.

$$\varepsilon_k^{\tau^+}(s) = \frac{4}{q^{\min}} \sqrt{\frac{2}{V_{k-1}^+(s)} \log\left(\frac{2S}{\delta_k}\right)}$$

$$\varepsilon_k^{\tau^-}(s) = \frac{4}{q^{\min}} \sqrt{\frac{2}{V_{k-1}^-(s)} \log\left(\frac{2S}{\delta_k}\right)} \quad (13)$$

We estimate the conditional probabilities of each event type conditioned on whether they are a positive or negative event. We use $p_s(l)$ to represent the probability of observing a type $l$ event in state $s$, which may or may not be accepted. Then, the conditional probabilities can be defined as $p_s^+(l) = p_s(i = l|i \in B^+(s))$ and $p_s^-(l) = p_s(i = l|i \in B^-(s))$. To estimate these conditional probabilities, we use the following estimator with a corresponding $\ell_1$ confidence bound, as was used

in (Auer et al., 2008; Gao & Zhou, 2024).

$$\hat{p}^+_{k,s}(l) = \frac{V_{k-1}(s,l)}{V^+_{k-1}(s)}$$

$$\hat{p}^-_{k,s}(l) = \frac{V_{k-1}(s,l)}{V^-_{k-1}(s)}$$

The corresponding confidence bound for the $\ell_1$ norm, $\varepsilon^p_k(s)$ is given below. Note that unlike (Auer et al., 2008; Gao & Zhou, 2024), but similarly to (Weber et al., 2024), we are estimating probabilities of individual event types rather than transitions between states. Therefore, we simply use the union bound over the number of states, as done before, with a square root term for the number of event types in each state, $N+1$. Also note the coefficient of 2 rather than 14 as used in (Auer et al., 2008), as we are using a slower confidence schedule. The following $\ell_1$ bounds hold with probability of at least $1 - \frac{\delta}{S}$.

$$\varepsilon^{p^+}_k(s) = \sqrt{\frac{2(N^s+1)}{V^+_{k-1}(s)} \log\left(\frac{2S}{\delta_k}\right)}$$

$$\varepsilon^{p^-}_k(s) = \sqrt{\frac{2(N^c+1)}{V^-_{k-1}(s)} \log\left(\frac{2S}{\delta_k}\right)} \tag{14}$$

Then, we define the confidence set $\mathcal{D}_k$ below

$$\mathcal{D}_k = \{D | \left| \frac{1}{\Lambda_D(s) + \eta_D(s)} - \hat{\tau}^+_k(s) \right| \le \varepsilon^{\tau^+}_k(s) \quad \forall s\}$$

$$\cap \{D | \left| \frac{1}{M_D(s) + \gamma_D(s)} - \hat{\tau}^-_k(s) \right| \le \varepsilon^{\tau^-}_k(s) \quad \forall s\}$$

$$\cap \{D | \left\| p^+_{s,D} - \hat{p}^+_{k,s} \right\| \le \varepsilon^{p^+}_k(s) \quad \forall s\}$$

$$\cap \{D | \left\| p^-_{s,D} - \hat{p}^-_{k,s} \right\| \le \varepsilon^{p^-}_k(s) \quad \forall s\}$$

### B.2 FINDING AN OPTIMISTIC POLICY

Next, we describe how to find an optimistic policy. There are two principal challenges to be solved first. The first is enforcing Assumption 2, and the second is determining optimism around abandonment rates. The first can be solved by truncating the confidence set appropriately. The second challenge is solved by choosing the lowest abandonment rate and including an additional fictitious customer and server type each, in order to allow the agent a choice between upper and lower bounds.

We begin with defining the extended model $D^{ext}_k$. In addition to having different rates, this model has a larger action set containing two fictitious types corresponding to excess abandonments, indexed as $N^c + 1$ for server abandonments and $N^s + 1$ for customer abandonments. We first construct the abandonment rates, $\eta^{ext}_k(s)$ and $\gamma^{ext}(s)$. These are determined using the *lower* bound. This is because the choice between the upper and lower bound is handled by an extra customer type.

$$\eta^{ext}_k(s) = \begin{cases} \max\left( \frac{\max(\hat{p}^+_{k,s}(0) - \frac{1}{2}\varepsilon^{p^+}_k(s), 0)}{\hat{\tau}^+_k(s) + \varepsilon^{\tau^+}_k(s)}, \max_{s'>s} \eta^{ext}_k(s), q^{\min} \right) & s < 0 \\ 0 & s \ge 0 \end{cases} \tag{15}$$

$$\gamma^{ext}_k(s) = \begin{cases} 0 & s \le 0 \\ \max\left( \frac{\max(\hat{p}^-_{k,s}(0) - \frac{1}{2}\varepsilon^{p^-}_k(s), 0)}{\hat{\tau}^-_k(s) + \varepsilon^{\tau^-}_k(s)}, \max_{s'<s} \gamma^{ext}_k(s), q^{\min} \right) & s > 0 \end{cases} \tag{16}$$

For arrival rates, we denote the total customer arrival rate by $\Lambda^{ext}_k(s)$. This is derived by using the maximum possible value of $\Lambda(s)$ within the confidence set, plus the difference between the

maximum and maximum possible values of $\eta(s)$.

$$\Lambda_k^{ext}(s) = \min\left(\frac{1}{\max(\hat{\tau}_k^+(s) - \varepsilon_k^{\tau^+}(s), (\Lambda^{\max} + 1_{s<0}\eta^{\max})^{-1})},\right.$$

$$\left.\min_{s'<s} \Lambda_k^{ext}(s) + \eta^{ext}(s), \lambda^{\max} + 1_{s<0}\eta^{\max}\right) - \eta^{ext}(s) \tag{17}$$

Similarly, we can define the aggregate server arrival rate $M_k^{ext}(s)$

$$M_k^{ext}(s) = \min\left(\frac{1}{\max(\hat{\tau}_k^-(s) - \varepsilon_k^{\tau^-}(s), (M^{\max} + 1_{s>0}\gamma^{\max})^{-1})},\right.$$

$$\left.\min_{s'>s} M_k^{ext}(s) + \gamma^{ext}(s), \mu^{\max} + 1_{s>0}\gamma^{\max}\right) - \gamma^{ext}(s)$$

The reward probabilities for all types in the system are the same known quantities as in the full model. There is an extra customer type, indexed $N^c + 1$, with reward $r_{N^c+1}^c(s) = r^a(s)$. Also, there is an extra server type, indexed $N^s + 1$, with reward $r_{N^s+1}^s(s) = r^a(s)$.

Then, we present individual type probabilities in extended model below. We proceed by using the maximal event probability for the customer and server types with the highest rewards, and minimal event probabilities for those with the lowest rewards. We also ensure that the minimal abandonment rates, $\eta^{ext}(s)$ and $\gamma^{ext}(s)$ have the appropriate probability. The extra types corresponding to excess abandonments are treated similarly to other types, with an appropriate correction. The following probabilities maximize the reward from arrival types, while enforcing the minimum bound for abandonments. In particular, we put as much weight on the customer (server) type in the extended model with maximal reward, and reduce the probabilities for lower-reward types accordingly.

If $s < 0$, then for $l = 0$

$$p_{k,s}^{ext}(0|l \in B^+(s)) = \frac{\eta^{ext}(s)}{\eta^{ext}(s) + \lambda^{ext}(s)}$$

If $s > 0$, then for $l = 0$

$$p_{k,s}^{ext}(0|l \in B^-(s)) = \frac{\gamma^{ext}(s)}{\gamma^{ext}(s) + M_k^{ext}(s)}$$

We define the excess probability for positive and negative arrival types as

$$E_k^+(s) = \frac{1}{2}\varepsilon_k^{p^+}(s) - 1_{s<0}\max(0, p_{k,s}^{ext} - p_{k,s}(0))$$

$$E_k^-(s) = \frac{1}{2}\varepsilon_k^{p^-}(s) - 1_{s>0}\max(0, p_{k,s}^{ext} - p_{k,s}(0))$$

For all $0 < l \leq N^c$

$$p_{k,s}^{ext}(l|l \in B^+(s)) = \hat{p}_{k,s}^+(l) + 1_{l=M^{c,ext}(s)}E_k^+(s)$$

$$- \max(0, \frac{1}{2}\varepsilon_k^{p^+}(s) - \sum_{m \notin A^{c,ext}(s,l)\cup\{l,0\}} \hat{p}_{k,s}^+(m)$$

Finally, for $l = N^c + 1$

$$p_{k,s}^{ext}(N^c + 1|l \in B^+(s))) = \max\left(1, p_{k,s}^+(0), \hat{p}_{k,s}^+(l) + 1_{l=M^{c,ext}(s)}E_k^+(s)\right.$$

$$\left.- \max\left(p_{k,s}^+(0), \frac{1}{2}\varepsilon_k^{p^+}(s) - \sum_{m \notin A^{c,ext}(s,l)\cup\{l,0\}} \hat{p}_{k,s}^+(m)\right)\right)$$

$$- p_{k,s}^+(0)$$

Then, for all customer extended arrival types $i$

$$\lambda_{i,k}^{ext}(s) = p_{k,s}^{ext}(l=i|l \in B^+(s))[\Lambda_k^{ext}(s) + \eta_k^{ext}(s)]$$

Likewise, define $A^{s,ext}(s,i)$ to be the set of all extended server types $l$ such that $r^s(l) > r^s(i)$ or $r^s(l) = r^s(i)$ and $l < i$. Then, for all $0 > -l \geq -N^s$

$$p_{k,s}^{ext}(l|l \in B^-(s)) = \hat{p}_{k,s}^-(l) + 1_{l=M^{s,ext}(s)}E_k^-(s)$$

$$- \max(0, \frac{1}{2}\varepsilon_k^{p^-}(s) - \sum_{m \notin A^{s,ext}(s,l) \cup \{l,0\}} \hat{p}_{k,s}^+(-m)$$

Finally, for $l = -(N^s+1)$

$$p_{k,s}^{ext}(-(N^s+1)|l \in B^-(s))) = \max\Bigg(1, p_{k,s}^-(0), \hat{p}_{k,s}^-(l) + 1_{l=M^{s,ext}(s)}E_k^-(s)$$

$$- \max\bigg(p_{k,s}^-(0), \frac{1}{2}\varepsilon_k^{p^-}(s) - \sum_{m \notin A^{s,ext}(s,l) \cup \{l,0\}} \hat{p}_{k,s}^-(-m)\bigg)\Bigg)$$

$$- p_{k,s}^-(0)$$

Then, for all customer extended arrival types $i$

$$\mu_{j,k}^{ext}(s) = p_{k,s}^{ext}(l=j|l \in B^-(s))[M_k^{ext}(s) + \gamma_k^{ext}(s)]$$

The next results establish optimism of the extended model with respect to true model. These results are less direct than in prior work (Auer et al., 2008; Gao & Zhou, 2024), as the set of possible parameters is truncated to enforce Assumption 2, and a priori upper bounds for customer and server rates are found. The next result establishes that both the maximal event rates are greater than that of the true model. This will be used to show that for the optimal policy of the true model, a policy for the extended model with identical rates and greater reward in each state can be found.

**Lemma 5.** *Assume the true model is within the confidence set $\mathcal{D}_k$. Then for all states $s$*

$$\Lambda_k^{ext}(s) + \eta_k^{ext}(s) \geq \Lambda(s) + \eta(s) \tag{18}$$

$$M_k^{ext}(s) + \gamma_k^{ext}(s) \geq M(s) + \gamma(s) \tag{19}$$

$$\eta_k^{ext}(s) \leq \eta(s) \tag{20}$$

$$\gamma_k^{ext}(s) \leq \gamma(s)$$

*Furthermore, the extended model fulfills Assumption 2 and Assumption 3.*

*Proof.* We proceed by showing that these properties hold for any model $D$ within the confidence set that fulfills assumptions 1 and 3. In order to show (20), we first note that for all $D$ within $\mathcal{D}_k$ that fulfills both assumptions, and for all $s > 0$

$$\gamma_D(s) \geq \max\left(\frac{\max(\hat{p}_{k,s}^-(0) - \frac{1}{2}\varepsilon_k^{p^-}(s), 0)}{\hat{\tau}_k^-(s) + \varepsilon_k^{\tau^-}(s)}, q^{\min}\right)$$

Then, combining these and (16) implies that $\gamma_D(s) < \gamma_k^{ext}(s)$ if there exists some $s' > s$ such that $\gamma_k^{ext}(s') = \gamma_k^{ext}(s)$. Therefore, to derive a contradiction we consider the highest state $s'' > s$ such that $\gamma_k^{ext}(s'') = \gamma_k^{ext}(s)$. We know that in this case, $\gamma_k^{ext}(s'') \leq \gamma_D(s'')$, since otherwise there would be an even lower state with the same value. Then, if $\gamma_k^{ext}(s) > \gamma_D(s)$, we have $\gamma_D(s'') > \gamma_D(s)$, which contradictions Assumption 1. Therefore $\gamma_k^{ext}(s) \leq \gamma_D(s)$. A substantially identical argument proves (20) as well.

We then use a similar argument to show (19). We note that

$$\Lambda_D(s) + \eta_D(s) \leq \min\left(\frac{\sum_{l=1_{s\geq 0}}^{N^c} \hat{p}_k(s,l) + \frac{1}{2}\varepsilon_{p,k}(s)}{\max(\tau_k^+(s) - \varepsilon_k^{\tau^+}(s), (\Lambda^{\max} + 1_{s<0}\eta^{\max})^{-1})}, \Lambda^{\max} + 1_{s<0}\eta^{\max}\right)$$

Combining (17) and (15), either the following holds

$$\Lambda_k^{ext}(s) + \eta_k^{ext}(s) = \min\left(\frac{\sum_{l=1_{s\geq 0}}^{N^c} \hat{p}_k(s,l) + \frac{1}{2}\varepsilon_{p,k}(s)}{\max(\tau_k^+(s) - \varepsilon_k^{\tau^+}(s), (\Lambda^{\max} + 1_{s<0}\eta^{\max})^{-1})}, \Lambda^{\max} + 1_{s<0}\eta^{\max}\right)$$

or for some $s' < s$, $\Lambda_k^{ext}(s') + \eta_k^{ext}(s') = \Lambda_k^{ext}(s) + \eta_k^{ext}(s)$. In the first case, it follows that $\Lambda_k^{ext}(s) + \eta_k^{ext}(s) \geq \Lambda_D(s) + \eta_D(s)$. In the second case, we consider the lowest state $s''$ such that $\lambda_k^{ext}(s'') + \eta_k^{ext}(s'') = \Lambda_k^{ext}(s) + \eta_k^{ext}(s)$. We know that $\lambda_k^{ext}(s'') + \eta_k^{ext}(s'') \geq \Lambda_D(s'') + \eta_D(s'')$, since it must fall under the first case. Therefore, since $D$ fulfills Assumption 2

$$\begin{aligned}\Lambda_k^{ext}(s) + \eta_k^{ext}(s) &= \Lambda_k^{ext}(s'') + \eta_k^{ext}(s'') \\ &\geq \Lambda_D(s'') + \eta_D(s'') \\ &\geq \Lambda_D(s) + \eta_D(s)\end{aligned}$$

This completes the proof. A qualitatively similar proof shows that $M_k^{ext}(s) + \gamma_k^{ext}(s) \geq M_D(s) + \gamma_D(s)$.

Since $\Lambda_k^{ext}(s) \leq \lambda_k^{ext}(s')$ for all $s' > s$, $M_k^{ext}(s) \leq M_k^{ext}(s')$ for all $s' < s$, $\gamma_k^{ext}(s) \geq \gamma_k^{ext}(s')$ for all $s' < s$, and $\eta_k^{ext}(s) \geq \eta_k^{ext}(s')$ for all $s' > s$, we know that Assumption 2 is fulfilled. This concludes the proof. $\square$

The next result establishes a bound between the behavior of the model under any policy is close to that of the extended model. Policies that are defined on the extended action set can be mapped to policies under the true action set that simply ignores the extra customer and server types $N^c + 1$ and $N^s + 1$. We represent this mapping by $\Psi$. Where $a$ is the action taken with probability 1 in state $s$, let $a' = (a_1\backslash\{N^c + 1\}, a_2\backslash\{N^s + 1\})$. Then, let $\Psi(\pi)(s, a') = 1$ and $\Psi(\pi)(s, a'') = 0$ for all $a'' \neq a$.

**Lemma 6.** *Assume the true model is within the confidence set $\mathcal{D}_k$.*

*Then under any deterministic policy $\pi$ and for any state $s$ the following bounds hold.*

$$\begin{aligned}&\left|(\lambda_\pi^{ext}(s) + \eta^{ext}(s)) - (\lambda_{\Psi(\pi)} + \eta(s))\right| \\ &\quad \leq 4(\Lambda^{\max} + 1_{s<0}\eta^{\max})(\Lambda(s) + \eta(s))\varepsilon_k^{\tau^+}(s) + (\Lambda^{\max} + 1_{s<0}\eta^{\max})\varepsilon^{p^+}(s) \\ &\left|(\mu_\pi^{ext}(s) + \gamma^{ext}(s)) - (\mu_{\Psi(\pi)} + \gamma(s))\right| \\ &\quad \leq 4(M^{\max} + 1_{s>0}\gamma^{\max})(M(s) + \gamma(s))\varepsilon_k^{\tau^+}(s) + (M^{\max} + 1_{s>0}\gamma^{\max})\varepsilon^{p^+}(s) \quad (21)\end{aligned}$$

*Proof.* We bound $\left|(\lambda_\pi^{ext}(s) + \eta^{ext}(s)) - (\lambda_\pi + \eta(s))\right|$ for each state $s$. Let $a$ be the action chosen in state $s$. We temporarily use the notation

$$\bar{q}^+(s) = \frac{1}{\max(\hat{\tau}_k^+(s) - \varepsilon_k^{\tau^+}(s), (\lambda^{\max} + \eta^{\max})^{-1})}$$

We consider two cases, one in which the extra abandonment type $N^c + 1$ is selected in state $s$, and another when it is not. Then, we note that that $p_s^+$ is within $\varepsilon^{p^+}(s)$ of the following vector with respect to the $\ell_1$ norm

$$\begin{pmatrix} p_{k,s}^{ext}(l = 0 \vee l = N^c + 1 | l \in B^+(s)) \\ p_{k,s}^{ext}(l = 1 | l \in B^+(s)) \\ \dots \\ p_{k,s}^{ext}(l = N^c | l \in B^+(s)) \end{pmatrix}$$

It follows from this that

$$|(\lambda_\pi^{ext}(s) + \eta^{ext}(s)) - (\lambda_{\Psi(\pi)} + \eta(s))|$$

$$\leq \left| \bar{q}^+(s) \sum_{i \in a_1 \cup \{0\}} p_{k,s}^{ext}(l = i | l \in B^+(s)) - (\Lambda + \eta(s)) \sum_{i \in a_1 \cup \{0\}} p_{k,s}^+(i) \right|$$

$$\leq |\bar{q}^+(s) - (\Lambda + \eta(s))| + (\Lambda^{max} + 1_{s<0}\eta^{max}) \left| \sum_{i \in a_1 \cup \{0\}} p_s^{ext}(l = i | l \in B^+(s)) - p_s^+(i) \right|$$

$$\leq 2(\Lambda^{max} + 1_{s<0}\eta^{max})(\Lambda(s) + \eta(s))\varepsilon_k^{\tau^+}(s) + \frac{1}{2}(\Lambda^{max} + 1_{s<0}\eta^{max})\varepsilon^{p^+}(s)$$

Then, in the case that $N^c + 1 \notin a_1$ and $s < 0$

$$|(\lambda_\pi^{ext}(s) + \eta^{ext}(s)) - (\lambda_{\Psi(\pi)} + \eta(s))|$$

$$\leq \left| \bar{q}^+(s) \sum_{i \in a_1} p_{k,s}^{ext}(l = i | l \in B^+(s)) - (\Lambda + \eta(s)) \sum_{i \in a_1} p_{k,s}^+(i) \right| + |\eta^{ext}(s) - \eta(s)|$$

$$\leq 2 \left| \bar{q}^+(s) - (\Lambda + \eta(s)) \right| + (\Lambda^{max} + 1_{s<0}\eta^{max}) \left| \sum_{i \in a_1 \cup \{0\}} p_s^{ext}(l = i | l \in B^+(s)) - p_s^+(i) \right|$$

$$+ (\Lambda^{max} + 1_{s<0}\eta^{max})\varepsilon^{p^+}(s)$$

$$\leq 4(\Lambda^{max} + 1_{s<0}\eta^{max})(\Lambda(s) + \eta(s))\varepsilon_k^{\tau^+}(s) + (\Lambda^{max} + 1_{s<0}\eta^{max})\varepsilon^{p^+}(s)$$

A qualitatively identical argument establishes (21). This completes the proof $\qquad\square$

Finally, the next proposition establishes optimism of the extended model with regards to the true model. In particular, we proceed by finding a corresponding policy for the extended model with identical rates and greater rewards in each state. The existence of a policy with equal rates, contingent on the true model being found within the confidence set, is established by Lemma 5. Then, the fact that the extended model gives more weight to the conditional probability of higher-reward types enables us to show that there exists such a policy with a higher reward in each state.

**Proposition 7.** *If the true model lies within the confidence set $\mathcal{D}_k$, then the maximal gain of the true model is less than that of the extended model.*

*Proof.* We use $D$ to represent the true model. We then construct a version the true model with an identical action set to the extended model, which we denote by $D'$. All rates are equal, with the exception of adding in the fictitious types for abandonments, and lowering the abandonment rates accordingly.

$$\lambda_{N^c+1, D'}(s) = \eta(s) - \eta_k^{ext}(s)$$
$$\eta_{D'}(s) = \eta_k^{ext}(s)$$
$$\mu_{N^s+1, D'}(s) = \gamma(s) - \gamma_k^{ext}(s)$$
$$\gamma_{D'}(s) = \gamma_k^{ext}(s)$$

Since it is known that $q^{min} \leq \gamma_k^{ext}(s) \leq \gamma(s)$ and $q^{min} \leq \eta_k^{ext}(s) \leq \eta(s)$, all rates are non-negative and $D'$ fulfills Assumption 3. Since the true model is assumed to fulfill Assumption 1, $D'$ must fulfill Assumption 2. It is clear that the maximal gain of $D'$ is at least as much as the maximal gain of the true model, since for any policy $\pi$ of the true model, there exists a policy $\pi'$ in the extended model where $N^c+1$ and $N^s+1$ are always accepted, and therefore have identical rates and rewards.

Considering the model $D'$, we have for all $i \leq N^c$ and $j \leq N^s$

$$\lambda_{i, D'}(s) = p_{s, D}(l = i | l \in B^+(s))(\Lambda_M(s) + \eta_M(s))$$
$$\mu_{j, D'}(s) = p_{s, D}(l = -j | l \in B^-(s))(M_M(s) + \gamma_M(s))$$

Furthermore, for the types corresponding to excesses abandonments

$$\lambda_{N^c+1,D'}(s) = p_{s,D}(l=0|l \in B^+(s))(\Lambda_M(s) + \eta_M(s)) - \eta_k^{ext}(s)$$

$$\mu_{N^s+1,D'}(s) = p_{s,D}(l=0|l \in B^-(s))(M_M(s) + \gamma_M(s)) - \gamma_k^{ext}(s)$$

First, we must show that for any state $s$ and customer type $i$, $p_{k,s}^{ext}(l \in A^{c,ext}(s,i)|l \in B^+) \geq p_{s,D'}(l \in A^{c,ext}(s,i)|l \in B^+)$. Since $p_{k,s}^{ext}(l \in A^{c,ext}(s,i)|l \in B^+)$ is the maximal value in the $\ell_1$ confidence set for all threshold types $i$, we have

$$p_{k,s}^{ext}(l \in A^{c,ext}(s,i)|l \in B^+) \geq p_{s,D'}(l \in A^{c,ext}(s,i)|l \in B^+) \tag{22}$$

Likewise, the same argument with regards to events in $B^+$, for any server type $j$

$$p_{k,s}^{ext}(l \in A^{s,ext}(s,j)|-l \in B^-) \geq p_{s,D'}(l \in A^{s,ext}(s,j)|-l \in B^-) \tag{23}$$

Next, we consider an optimal policy $\pi_d^*$ of $D'$ with the properties given in Corollary 1. Then, define the following policy $\pi^*$, according to Lemma 2, with the following rates for each state $s$

$$\lambda_{\pi^*,i,D'}(s) = \min\left(\lambda_{i,D'}(s), \lambda_{\pi_d^*,D'}(s) - \sum_{l \in A^{c,ext}(s,i)} \lambda_{\pi^*,l,D'}(s)\right)$$

$$\mu_{\pi^*,j,D'}(s) = \min\left(\mu_{j,D'}(s), \mu_{\pi_d^*,D'}(s) - \sum_{l \in A^{s,ext}(s,j)} \mu_{\pi^*,l,D'}(s)\right)$$

It can be then shown algebraically that the rates, rewards, and therefore gain are equal when $\pi^*$ and $\pi_d^*$ are both chosen, since the rates must only differ with respect to the customer and service types with rewards equal to the threshold value (if it exists), respectively. Furthermore, it also follows that there exists a customer type $i$ and server type $j$ such that if $l \in A^{c,ext}(s,i)$ then $\lambda_{\pi,i,D'}(s) = \lambda_{i,D'}(s)$ and if $l \in A^{s,ext}(s,j)$ then $\mu_{\pi,j,D'}(s) = \mu_{j,D'}(s)$. Furthermore, if $l \neq i$ and $\notin A^{c,ext}(s,i)$ then $\lambda_{\pi,i,D'}(s) = 0$ and if $l \neq j$ and $l \notin A^{s,ext}(s,j)$ then $\mu_{\pi,j,D'}(s) = 0$. We will call a type with this property a threshold type, since the next policy defined has types of the same property.

Likewise, also using Lemma 2, we can find another policy $\pi'$ for the extended model, with the same rates.

$$\lambda_{\pi',i}^{ext}(s) = \min\left(\lambda_{i,D'}(s), \lambda_{\pi_d^*}^{ext}(s) - \sum_{l \in A^{c,ext}(s,i)} \lambda_{\pi^*,l}^{ext}(s)\right)$$

$$\mu_{\pi',j}^{ext}(s) = \min\left(\mu_{j,D'}(s), \mu_{\pi_d^*,D'}(s) - \sum_{l \in A^{s,ext}(s,j)} \mu_{\pi^*,l,D'}(s)\right)$$

Then, let $i'$ and $j'$ be the customer and server threshold types for $\pi'$. Since $A^{c,ext}(s,i)$ and $A^{s,ext}$ represent a lexicographical ordering, they also define total orderings.

Then, (22) implies that for each state the sequence of $\lambda_{l,\pi'}^{ext}(s)$ majorizes $\lambda_{l,\pi^*,D'}(s)$ over $l$ with regards to order induced by $A^{c,ext}(s,\cdot)$. Likewise, by (23) we know $\mu_{l,\pi'}^{ext}(s)$ majorizes $\mu_{l,\pi^*,D'}(s)$ over the $l$ with regards to the order induced by $A^{s,ext}(s,\cdot)$. Then we note that $r_i^c(s)$ and $r_j^s(s)$ are decreasing over $i$ and $j$ with respect to the ordering defined by $A^{c,ext}(s,\cdot)$ and $A^{s,ext}(s,\cdot)$, respectively. It is clear by Proposition B.7 in (Marshall et al., 1979), a consequence of the Hardy-Littlewood-Polya inequality, that $\sum_l r_l^c \lambda_{\pi',l}^{ext}(s) \geq \sum_l r_l^c \lambda_{\pi^*,l,D'}(s)$ and $\sum_l r_l^s \mu_{\pi',l}^{ext}(s) \geq \sum_l r_l^s \mu_{\pi^*,l,D'}(s)$.

Since all aggregate event rates are identical, and the abandonment rates are unchanged, we have $g_{\pi'}^{ext} \geq g_{\pi^*,D'}$. This completes the proof. □

## C  REGRET ANALYSIS

In this section, we will primarily use the adjusted regret, from (Gao & Zhou, 2024). This replaces the sojourn time within each time-step with the average sojourn time, and the per-step reward with

the average reward. The total adjusted regret from steps within episode $k$ is equal to

$$\bar{\Delta}_k = \sum_{t=V_{k-1}+1}^{V_{k-1}+v_k} \bar{\tau}(s(t))(g^* - \bar{r}_{\pi_k}(s))$$

In particular, this allows us to simplify the comparison of rewards in the true model and the extended model under a given policy, and allows for bounds without needing to consider tails of the sojourn time. Later on, we will show that it is equal in expectation to the regret, defined for episode $k$ below, where $r(t)$ is the observed reward at step $t$

$$\Delta_k = \sum_{t=V_{k-1}+1}^{V_{k-1}+v_k} = \tau_t(g^* - r(t))$$

Next, we present a result that bounds the total number of episodes up to step $T$. This will be used later to deal with constant factors in the episodic regret, such as the regret from out-of-confidence episodes and imbalanced steps.

**Lemma 7.** *(Auer et al., 2008) The total number of episodes $K$ before time step $T > S$ is bounded above*

$$K \le S \log_2\left(\frac{8T}{S}\right)$$

*Proof.* This follows from a straightforward adjustment of Proposition 18 in (Auer et al., 2008), from noting that we explore over states instead of state-action pairs. □

Next, we consider both out of confidence episodes and regret from imbalanced states. We define an imbalanced state as any state $n$ in episode $k$ such that one of the following holds

$$V_{k-1}^+(s) \le \frac{1}{2}p^+(s)V_{k-1}(s)$$

$$V_{k-1}^-(s) \le \frac{1}{2}p^-(s)V_{k-1}(s)$$

Then, we use $F_k^{imb}$ to represent the set of imbalanced states in episode $k$, and $F_k^{bal} = [\underline{s}, \bar{s}] \backslash F_k^{imb}$ represent the set of balanced states. In particular, the notion of balanced states is necessary to ensure square-root regret bounds, by bounding the probability that $V_{k-1}^+(s)$ and $V_{k-1}^-(s)$ are much lower than their respective probabilities multiplied by $V_{k-1}(s)$. An imbalanced step is any step $t$ in episode $k$ such that $s(t) \in F_k^{imb}$, and we present a lemma bounding the regret from imbalanced steps below.

**Lemma 8.** *The cumulative adjusted regret from imbalanced steps up to episode $K$ is upper bounded*

$$\mathcal{R}_K^{imb} \le 8\kappa^2 S \frac{q^{\max} + 1}{q^{\min}} K$$

*Proof.* By the Azuma-Hoeffding inequality, we have the following

$$P\left(V_{k-1}^+(s) \le \frac{1}{2}p^+(s)V_{k-1}(s)\right) \le \exp\left(-\frac{1}{2}V_{k-1}(s)(p^+(s))^2\right)$$

$$\le \exp\left(\frac{-V_{k-1}(s)}{2\kappa^2}\right)$$

$$P\left(V_{k-1}^-(s) \le \frac{1}{2}p^-(s)V_{k-1}(s)\right) \le \exp\left(-\frac{1}{2}V_{k-1}(s)(p^-(s))^2\right)$$

$$\le \exp\left(\frac{-V_{k-1}(s)}{2\kappa^2}\right)$$

Therefore, applying a union bound over both positive and negative events we get

$$P(s \in F_k^{imb}|V_{k-1}(s) = v') \leq 2\exp\left(\frac{-v'}{2\kappa^2}\right)$$

Then, noting that the per-step regret is upper bounded by $\frac{2(q^{\max}+1)}{q^{\min}}$, $v_k(s) \leq V_{k-1}(s)$, and for any non-negative $u$, $0 \leq ue^{-u} \leq \frac{1}{e}$

$$\mathbb{E}[\Delta_k^{imb}] \leq 2\frac{q^{\max}+1}{q^{\min}}\sum_{s=\underline{s}}^{\bar{s}}\mathbb{E}[v_k(s)1_{s \in F_k^{imb}}]$$

$$\leq 2\frac{q^{\max}+1}{q^{\min}}\sum_{s=\underline{s}}^{\bar{s}}\mathbb{E}[V_{k-1}(s)1_{s \in F_k^{imb}}]$$

$$\leq 2\frac{q^{\max}+1}{q^{\min}}\sum_{s=\underline{s}}^{\bar{s}}\sum_{v'=0}^{\infty}v'P(V_{k-1}(s) = v')P(s \in F_k^{imb}|V_{k-1}(s) = v')$$

$$\leq 2\frac{q^{\max}+1}{q^{\min}}\sum_{s=\underline{s}}^{\bar{s}}\sum_{v'=0}^{\infty}2v'P(V_{k-1}(s) = v')\exp\left(\frac{-v'}{2\kappa^2}\right)$$

$$\leq 8\frac{q^{\max}+1}{q^{\min}}\sum_{s=\underline{s}}^{\bar{s}}\sum_{v'=0}^{\infty}\frac{1}{e}\kappa^2 P(V_{k-1}(s) = v')$$

$$\leq 8S\kappa^2\frac{q^{\max}+1}{q^{\min}}$$

□

Next, as necessary in UCL-inspired algorithms, we present a bound on the regret from episodes in which the true model falls outside the confidence set $\mathcal{D}_k$. This follows directly from using the maximal regret and multiplying by the probability that the true model does not fall within the given confidence set.

**Lemma 9.** *The cumulative adjusted regret for out of bound episodes up to episode $K$, $\bar{\mathcal{R}}_K^{out}$, is upper bounded in expectation*

$$\mathbb{E}[\bar{\mathcal{R}}_K^{out}] \leq 8\frac{q^{\max}+1}{q^{\min}}\delta K$$

*Proof.* We begin by finding an upper bound for $P[M \notin \mathcal{D}_k]$. With a union bound over the state space and individual parameters, we have

$$P[M \notin \mathcal{D}_k] \leq \sum_{s=\underline{s}}^{\bar{s}}(P[|\tau^+(s) - \hat{\tau}_k^+(s)| > \varepsilon_k^{\tau^+}(s)] + P[|\tau^-(s) - \hat{\tau}_k^-(s)| > \varepsilon_k^{\tau^-}(s)]$$

$$+ P[\|p_s^+ - \hat{p}_{k,s}^+\|_1 > \varepsilon_k^{p^+}(s)]) + P[\|p_s^- - \hat{p}_{k,s}^-\|_1 > \varepsilon_k^{p^-}(s)])$$

$$\leq S4\frac{\delta_k}{S} \leq 4\delta_k \leq \frac{4}{V_{k-1}}\delta$$

Then, bounds for $\bar{\mathcal{R}}_K^{out}$ follow easily from this.

$$\mathbb{E}[\mathcal{R}_K^{out}] = \mathbb{E}[\sum_{k=0}^{K}1_{M \notin \mathcal{D}_k}\bar{\Delta}_k]$$

$$\leq \sum_{k=0}^{K}2\frac{q^{\max}+1}{q^{\min}}v_k\mathbb{E}[1_{M \notin \mathcal{D}_k}]$$

$$= \sum_{k=0}^{K} 2\frac{q^{\max}+1}{q^{\min}} v_k P[M \notin \mathcal{D}_k]$$

$$\leq 2\frac{q^{\max}+1}{q^{\min}} \sum_{k=0}^{K} \frac{4v_k}{V_{k-1}}\delta$$

$$\leq 8\frac{q^{\max}+1}{q^{\min}}\delta K$$

$\square$

Next, we proceed with the core of the regret proof, the regret in balanced steps when the model falls within the confidence set. We begin by establishing the following results, which establish that both the event rates and the rewards (under the optimal policy) in the extended model are close to that of the true model. The main point of interest is that the inner denominator within the confidence bounds is either $V_{k-1}^+(s)$ or $V_{k-1}^-(s)$, rather than $V_{k-1}(s)$. This can be mended by the fact that $V_{k-1}^+(s), V_{k-1}^-(s) \geq \frac{\kappa}{2}V_{k-1}(s)$ in balanced steps, and this can be plugged in to establish results in terms of $V_{k-1}(s)$.

**Lemma 10.** *In any episode $k$ where the true model falls within the confidence set, we have for any balanced state $s$*

$$\left|(\lambda_{\hat{\pi}_k}^{ext}(s) + \eta^{ext}(s)) - (\lambda_{\pi_k} + \eta(s))\right| + \left|(\mu_{\hat{\pi}_k}^{ext}(s) + \gamma^{ext}(s)) - (\mu_{\pi_k} + \gamma(s))\right|$$

$$\leq \left(32\kappa q^{\max} + 2\sqrt{\kappa(N+1)}q^{\max}\right)\sqrt{\frac{1}{V_{k-1}(s)}\log\left(\frac{2S}{\delta_k}\right)}$$

*Proof.* First, we note that for any balanced state $s$, both $V_{k-1}^+(s) \geq 1$ and $V_{k-1}^-(s) \geq 1$. Then, combining (13) and (14), with the fact that $V_{k-1}^+(s) \geq \frac{1}{2}p^+(s)V_{k-1}(s)$ and $V_{k-1}^-(s) \geq \frac{1}{2}p^-(s)V_{k-1}(s)$, we have

$$\varepsilon_k^{\tau^+}(s) \leq \frac{4}{q^{\min}}\sqrt{\frac{4}{p^+(s)V_{k-1}(s)}\log\left(\frac{2S}{\delta_k}\right)}$$

$$\varepsilon_k^{\tau^-}(s) \leq \frac{4}{q^{\min}}\sqrt{\frac{4}{p^-(s)V_{k-1}(s)}\log\left(\frac{2S}{\delta_k}\right)}$$

$$\varepsilon_k^{p^+}(s) \leq \sqrt{\frac{4(N^s+1)}{p^+(s)V_{k-1}(s)}\log\left(\frac{2S}{\delta_k}\right)}$$

$$\varepsilon_k^{p^-}(s) \leq \sqrt{\frac{4(N^c+1)}{p^-(s)V_{k-1}(s)}\log\left(\frac{2S}{\delta_k}\right)}$$

Therefore

$$\varepsilon_k^{\tau^+}(s) \leq \frac{8}{q^{\min}}\sqrt{\frac{\Lambda(s) + M(s) + \eta(s) + \gamma(s)}{(\Lambda(s) + \eta(s))V_{k-1}(s)}\log\left(\frac{2S}{\delta_k}\right)}$$

$$\varepsilon_k^{\tau^-}(s) \leq \frac{8}{q^{\min}}\sqrt{\frac{\Lambda(s) + M(s) + \eta(s) + \gamma(s)}{(M(s) + \gamma(s))V_{k-1}(s)}\log\left(\frac{2S}{\delta_k}\right)}$$

$$\varepsilon_k^{p^+}(s), \varepsilon_k^{p^-}(s) \leq 2\sqrt{\frac{\kappa(N+1)}{V_{k-1}(s)}\log\left(\frac{2S}{\delta_k}\right)}$$

By Lemma 6, we have

$$\left|(\lambda_{\hat{\pi}_k}^{ext}(s) + \eta^{ext}(s)) - (\lambda_{\pi_k} + \eta(s))\right|$$

$$\leq 4(\Lambda^{\max} + 1_{s<0}\eta^{\max})(\Lambda(s) + \eta(s))\varepsilon_k^{\tau^+}(s) + (\Lambda^{\max} + 1_{s<0}\eta^{\max})\varepsilon^{p^+}$$

$$\left|(\mu_{\hat{\pi}_k}^{ext}(s) + \gamma^{ext}(s)) - (\mu_{\pi_k} + \gamma(s))\right|$$

$$\leq 4(M^{\max} + 1_{s>0}\gamma^{\max})(M(s) + \gamma(s))\varepsilon_k^{\tau^+}(s) + (M^{\max} + 1_{s>0}\gamma^{\max})\varepsilon^{p^+}(s)$$

To illustrate our simplification of this, we focus on simplifying $(\Lambda^{\max} + 1_{s<0}\eta^{\max})(\Lambda(s) + \eta(s))\varepsilon_k^{\tau^+}(s)$. This can be done by

$$(\Lambda^{\max} + 1_{s<0}\eta^{\max})(\Lambda(s) + \eta(s))\varepsilon_k^{\tau^+}(s)$$

$$\leq (\Lambda^{\max} + 1_{s<0}\eta^{\max})(\Lambda(s) + \eta(s))\frac{8}{q^{\min}}\sqrt{\frac{\Lambda(s) + M(s) + \eta(s) + \gamma(s)}{(\Lambda(s) + \eta(s)))V_{k-1}(s)}\log\left(\frac{2S}{\delta_k}\right)}$$

$$= (\Lambda^{\max} + 1_{s<0}\eta^{\max})\frac{8}{q^{\min}}\sqrt{\frac{(\Lambda(s) + M(s) + \eta(s) + \gamma(s))(\Lambda(s) + \eta(s))}{V_{k-1}(s)}\log\left(\frac{2S}{\delta_k}\right)}$$

$$\leq 8(\Lambda^{\max} + 1_{s<0}\eta^{\max})\kappa\sqrt{\frac{1}{V_{k-1}(s)}\log\left(\frac{2S}{\delta_k}\right)}$$

Then after a few more straightforward simplification steps, we can complete the proof

$$\left|(\lambda_{\hat{\pi}_k}^{ext}(s) + \eta^{ext}(s)) - (\lambda_{\pi_k} + \eta(s))\right| + \left|(\mu_{\hat{\pi}_k}^{ext}(s) + \gamma^{ext}(s)) - (\mu_{\pi_k} + \gamma(s))\right|$$

$$\leq \left(32\kappa q^{\max} + 2\sqrt{\kappa(N+1)}q^{\max}\right)\sqrt{\frac{1}{V_{k-1}(s)}\log\left(\frac{2S}{\delta_k}\right)}$$

$\square$

**Lemma 11.** *Let episode $k$ be an episode such that the true model is within $\mathcal{D}_k$, and let $\bar{r}_{\hat{\pi}_k}^{ext}(s)$ is the mean reward in the extended model in state $s$ under policy $\hat{\pi}_k$. If $s \in F_k^{bal}$, we have*

$$\bar{r}_{\hat{\pi}_k}^{ext}(s) - \bar{r}_{\pi_k}(s) \leq \left(32\kappa q^{\max} + 4\sqrt{\kappa(N+1)}q^{\max}\right)\sqrt{\frac{1}{V_{k-1}(s)}\log\left(\frac{2S}{\delta_k}\right)}$$

*Proof.* Let $a$ be the action chosen in state $s$ under policy $\hat{\pi}_k$. Then, where

$$r^{+,ext}(s) = p_s^{ext}(m = 0|0 \in B^+(s))r^a + \sum_{i=1}^{N^c+1} 1_{i \in a_1}p_s^{ext}(m = i|m \in B^+(s))r_i^c$$

$$r^{-,ext}(s) = p_s^{ext}(m = 0|0 \in B^-(s))r^a + \sum_{j=1}^{N^s+1} 1_{j \in a_2}p_s^{ext}(m = -j|m \in B^-(s))r_i^s$$

$$\bar{r}_{\hat{\pi}_k}^{ext}(s) = r^h + (\Lambda^{ext}(s) + \eta^{ext}(s))r^{+,ext}(s) + (M^{ext}(s) + \gamma^{ext}(s))r^{-,ext}(s)$$

Then again, as in the proof of Lemma 6, we note that that $p_s^+$ is within $\varepsilon^{p^+}(s)$ of the following vector with respect to the $\ell_1$ norm

$$\begin{pmatrix} p_{k,s}^{ext}(l = 0 \vee l = N^c + 1|l \in B^+(s)) \\ p_{k,s}^{ext}(l = 1|l \in B^+(s)) \\ \cdots \\ p_{k,s}^{ext}(l = N^c|l \in B^+(s)) \end{pmatrix}$$

A similar bound exists for $p_s^-(\cdot)$ and a similar vector conditioned on $B^-(s)$, with a corresponding $\ell_1$ bound of $\varepsilon^{p^-}(s)$. It then follows from noting that all reward values are bounded within $[-1, 1]$ and applying the same logic in Lemma 6, but taking the $\ell_1$ norm of probabilities instead of the total difference

$$\bar{r}_{\hat{\pi}_k}^{ext}(s) - \bar{r}_{\pi_k}(s) \leq 2(\Lambda^{\max} + 1_{s<0}\eta^{\max})\varepsilon^{p^+}(s) + 2(M^{\max} + 1_{s>0}\gamma^{\max})\varepsilon^{p^-}(s)$$

$$+ 4(\Lambda^{\max} + 1_{s<0}\eta^{\max})^2\varepsilon^{\tau^+}(s) + 4(M^{\max} + 1_{s>0}\gamma^{\max})^2\varepsilon^{\tau^-}(s)$$

Then, applying a similar argument to the derivation in Lemma 10, we can complete the proof

$$\bar{r}_{\hat{\pi}_k}^{ext}(s) - \bar{r}_{\pi_k}(s) \leq \left(32\kappa q^{\max} + 4\sqrt{\kappa(N+1)}q^{\max}\right)\sqrt{\frac{1}{V_{k-1}(s)}\log\left(\frac{2S}{\delta_k}\right)}$$

$\square$

The next result gives a bound that will become the dominating regret term. The main step uses the gain-bias equations to expand $g^{ext}_{\pi_k} - \bar{r}^{ext}_{\pi_k}(s)$. Then, the two prior results can be used, along with the bias bounds found in Proposition 3. We also include a martingale difference sequence to account for the regret from the bias changes that occur when the policy changes.

**Lemma 12.** *For any episode $k$ such that the true model is within $\mathcal{D}_k$, the expected adjusted regret in balanced steps, $\mathbb{E}[\bar{\Delta}^{bal}_k]$, is bounded above. Using $(\Delta h)^{\max} = \frac{2}{\min(q^{\min},1)}S(q^{\max}+1)$ to denote the upper bound on the relative bias from Proposition 3, the following bound holds*

$$\mathbb{E}[\bar{\Delta}^{bal}_k] \leq \sum_{s \in F^{bal}_k} v_k(s)\left[\left(32\kappa^2 + 4\kappa^{3/2}\sqrt{N+1}\right)\sqrt{\frac{1}{V_{k-1}(s)}\log\left(\frac{2S}{\delta_k}\right)}((\Delta h)^{\max}+1)\right]$$

$$+ S(\Delta h)^{\max}$$

*Proof.* As is common in finding square-root regret bounds (Auer et al., 2008; Anselmi et al., 2022), we begin by decomposing the adjusted regret into different terms, corresponding between the difference in gain and the reward under the extended model, and the difference in expected rewards between the extended and true models. We use $g^{ext}$, $h^{ext}(s)$, and $\bar{r}^{ext}(s)$ to represent the gain, as well as the bias and mean reward at state $s$ in the extended model. We also use $F^{bal}_k = [\underline{s}, \bar{s}] \backslash F^{imb}_k$ to represent the set of balanced states.

$$\mathbb{E}[\bar{\Delta}^{bal}_k] = \mathbb{E}\left[\sum_s 1_{s \in F^{bal}_k} v_k(s)\bar{\tau}(s)(g^* - \bar{r}_{\pi_k}(s))\right]$$

$$\leq \mathbb{E}\left[\left[\sum_{s \in F^{bal}_k}\sum_s v_k(s)\bar{\tau}(s)(g^{ext}_{\hat{\pi}_k} - \bar{r}_{\pi_k}(s))\right]\right.$$

$$\leq \mathbb{E}\left[\sum_{s \in F^{bal}_k} v_k(s)\bar{\tau}(s)(g^{ext}_{\pi_k} - \bar{r}_{\pi_k}(s))\right]$$

$$= \mathbb{E}\left[\sum_{s \in F^{bal}_k} v_k(s)\bar{\tau}(s)(g^{ext}_{\hat{\pi}_k} - \bar{r}^{ext}_{\hat{\pi}_k}(s))\right]$$

$$+ \mathbb{E}\left[\sum_{s \in F^{bal}_k} v_k(s)\bar{\tau}(s)(\bar{r}^{ext}_{\hat{\pi}_k}(s) - \bar{r}_{\pi_k}(s))\right]$$

We find bounds on $\mathbb{E}[\sum_{s \in F^{bal}_k} v_k(s)\bar{\tau}(s)(g^{ext}_{\hat{\pi}_k} - \bar{r}^{ext}_{\hat{\pi}_k}(s))]$ first. We decompose this term again using the gain-bias equations.

$$\mathbb{E}\left[\sum_{s \in F^{bal}_k} v_k(s)\bar{\tau}(s)(g^{ext}_{\hat{\pi}_k} - \bar{r}^{ext}_{\hat{\pi}_k}(s))\right]$$

$$= \mathbb{E}\left[\sum_{s \in F^{bal}_k} v_k(s)\bar{\tau}(s)((\lambda^{ext}_{\hat{\pi}_k}(s) + \eta^{ext}(s))\Delta h^{ext}_{\hat{\pi}_k}(s) - (\mu^{ext}_{\hat{\pi}_k}(s) + \gamma^{ext}(s))\Delta h^{ext}_{\hat{\pi}_k}(s-1))\right]$$

$$= \mathbb{E}\left[\sum_{s \in F^{bal}_k}\left[v_k(s)\bar{\tau}(s)((\lambda^{ext}_{\hat{\pi}_k}(s) + \eta^{ext}(s)) - (\lambda_{\pi_k}(s) + \eta(s)))\Delta h^{ext}_{\hat{\pi}_k}(s)\right.\right.$$

$$\left.\left. - ((\mu^{ext}_{\hat{\pi}_k}(s) + \gamma^{ext}(s)) - (\lambda_{\pi_k}(s) + \eta(s)))\Delta h^{ext}_{\hat{\pi}_k}(s-1))\right]\right] \tag{24}$$

$$+ \mathbb{E}\left[\sum_{s \in F^{bal}_k} v_k(s)\bar{\tau}(s)((\lambda_{\pi_k}(s) + \eta(s))\Delta h^{ext}_{\hat{\pi}_k}(s) - (\mu_{\pi_k}(s) + \gamma(s))\Delta h^{ext}_{\hat{\pi}_k}(s-1))\right] \tag{25}$$

Firstly, in order to find bounds on (24), we show it is bounded when conditioned on any sequence of state observations. We use $(\Delta h)^{\max}$ to represent the bias bounds found in Proposition 3. Considering an arbitrary $v_k(\cdot)$ and applying the bounds established in Lemma 6, the following can be

established

$$\sum_{s \in F_k^{bal}} v_k(s)\bar{\tau}(s)\bigg[((\lambda_{\hat{\pi}_k}^{ext}(s) + \eta^{ext}(s)) - (\lambda_{\pi_k}(s) + \eta(s)))\Delta h_{\hat{\pi}_k}^{ext}(s)$$

$$- ((\mu_{\hat{\pi}_k}^{ext}(s) + \gamma^{ext}(s)) - (\mu_{\pi_k}(s) + \gamma(s)))\Delta h_{\hat{\pi}_k}^{ext}(s-1)\bigg]$$

$$\leq \sum_{s \in F_k^{bal}} v_k(s)\bar{\tau}(s)\bigg[\big|(\lambda_{\hat{\pi}_k}^{ext}(s) + \eta^{ext}(s)) - (\lambda_{\pi_k} + \eta(s))\big|(\Delta h)^{\max}$$

$$+ \big|(\mu_{\hat{\pi}_k}^{ext}(s) + \gamma^{ext}(s)) - (\mu_{\pi_k} + \gamma(s))\big|(\Delta h)^{\max}\bigg]$$

*(Applying bias bounds)*

$$\leq \sum_{s \in F_k^{bal}} v_k(s)\bar{\tau}(s)\bigg[\bigg(32\kappa q^{\max} + 4\sqrt{\kappa(N+1)}q^{\max}\bigg)\sqrt{\frac{1}{V_{k-1}(s)}\log\bigg(\frac{2S}{\delta_k}\bigg)}(\Delta h)^{\max}\bigg]$$

*(Applying Lemma 10)*

$$\leq \sum_{s \in F_k^{bal}} v_k(s)\bigg[\bigg(32\kappa^2 + 4\kappa^{3/2}\sqrt{N+1}\bigg)\sqrt{\frac{1}{V_{k-1}(s)}\log\bigg(\frac{2S}{\delta_k}\bigg)}(\Delta h)^{\max}\bigg]$$

*(Noting that $\bar{\tau}(s) \leq \dfrac{1}{q^{\min}}$)*

Next, we find bounds on (25). We begin by considering the following sequence of random variables, in order to apply a variation of the martingale difference trick originating in (Anselmi et al., 2022)

$$\Phi_t = \bar{\tau}(s(t))\bigg[(\lambda_{\pi_k}(s(t)) + \eta(s(t))1_{(s(t)+1 \in F_k^{bal})}h^{ext}(s(t)+1)$$

$$+ (\mu_{\pi_k}(s(t)) + \gamma(s(t)))1_{(s(t)-1 \in F_k^{bal})}h^{ext}(s(t)-1)\bigg]$$

$$- 1_{(s(t+1) \in F_k^{bal})}h^{ext}(s(t+1))$$

We note that $P(s(t+1) = s'|s(t) = s) = \bar{\tau}(s)q(s, s')$. Therefore, conditioning on the sequence of states up to time $t$

$$\mathbb{E}[\Phi_t|s(V_{k-1}+1), \ldots, s(t)] = \bar{\tau}(s(t))\bigg[(\lambda_{\pi_k}(s(t)) + \eta(s(t))1_{(s(t)+1 \in F_k^{bal})}h^{ext}(s(t)+1)$$

$$+(\mu_{\pi_k}(s(t)) + \gamma(s(t)))1_{(s(t)-1 \in F_k^{bal})}h^{ext}(s(t)-1)\bigg] - E[1_{(s(t+1) \in F_k^{bal})}h^{ext}(s(t+1)] = 0$$

Since all values of $q_{\pi_k}(\cdot, \cdot)$ and $h^{ext}(\cdot)$ are bounded, and since

$$\mathbb{E}\bigg[\sum_{t=V_{k-1}+1}^{v_k} \Phi_t\bigg] = 0$$

this is a martingale difference sequence, and by the optional stopping theorem

$$\mathbb{E}\bigg[\sum_{s \in F_k^{bal}} v_k(s)\bar{\tau}(s)((\lambda_{\pi_k}(s) + \eta(s))\Delta h_{\hat{\pi}_k}^{ext}(s) - (\mu_{\pi_k}(s) + \gamma(s))\Delta h_{\hat{\pi}_k}^{ext}(s-1))\bigg]$$

$$= \mathbb{E}\bigg[\sum_{t=V_{k-1}+1}^{N_k} \Phi_t + 1_{(s(V_{k-1}) \in F_k^{bal})}h^{ext}(s(v_k)) - 1_{(s(V_{k-1}+1) \in F_k^{bal})}h^{ext}(s(V_{k-1}+1))\bigg]$$

$$\leq S(\Delta h)^{max}$$

Thus

$$\mathbb{E}\left[\sum_s v_k(s)\bar{\tau}(s)(\tilde{g}^*_{\pi_k} - \tilde{r}_{\pi_k}(s))\right]$$

$$\leq \sum_{s \in F_k^{bal}} v_k(s)\left[\left(32\kappa^2 + 2\sqrt{\kappa(N+1)}\right)\sqrt{\frac{1}{V_{k-1}(s)}\log\left(\frac{2S}{\delta_k}\right)}(\Delta h)^{\max}\right] + S(\Delta h)^{\max}$$

Next, we find bounds on $\mathbb{E}[\sum_{s \in F_k^{bal}} v_k(s)\bar{\tau}(s)(\tilde{r}_{\pi_k}(s) - \bar{r}_{\pi_k}(s))]$. This follows directly from applying Lemma 11 and noting that $v_k(s) \leq V_{k-1}(s)$ for all balanced states.

$$\sum_{s \in F_k^{bal}} v_k(s)\bar{\tau}(s)(\tilde{r}_{\pi_k}(s) - \bar{r}_{\pi_k}(s))$$

$$\leq \bar{\tau}(s) \sum_{s \in F_k^{bal}} v_k(s)\left(32\kappa q^{\max} + 4\sqrt{\kappa(N+1)}q^{\max}\right)\sqrt{\frac{1}{V_{k-1}(s)}\log\left(\frac{2S}{\delta_k}\right)}$$

$$\square$$

Then, combining all the terms together, we have

$$\mathbb{E}[\bar{\Delta}_k^{bal}] \leq S(\Delta h)^{\max}$$

$$+ \sum_{s \in F_k^{bal}} v_k(s)\left[\left(32\kappa^2 + 2\kappa^{3/2}\sqrt{N+1}\right)\sqrt{\frac{1}{V_{k-1}(s)}\log\left(\frac{2S}{\delta_k}\right)}(\Delta h)^{\max}\right]$$

$$+ \sum_{s \in F_k^{bal}} v_k(s)\left[\left(32\kappa^2 + 4\kappa^{3/2}\sqrt{N+1}\right)\sqrt{\frac{1}{V_{k-1}(s)}\log\left(\frac{2S}{\delta_k}\right)}\right]$$

$$\leq \sum_{s \in F_k^{bal}} v_k(s)\left[\left(32\kappa^2 + 4\kappa^{3/2}\sqrt{N+1}\right)\sqrt{\frac{1}{V_{k-1}(s)}\log\left(\frac{2S}{\delta_k}\right)}((\Delta h)^{\max} + 1)\right]$$

$$+ S(\Delta h)^{\max}$$

Finally, we combine the regret terms in balanced, in-confidence steps with those in unbalanced and out-of-confidence steps. This gives the final square root bounds, with Lemma 12 establishing the square root term and lemmas 8 and 9 establishing an additional logarithmic term. Furthermore, we show that the regret and adjusted regret are equal in expectation by using a martingale difference sequence, and therefore the bounds apply to both.

**Proposition 5.** *The total expected regret up to time $T$ is upper bounded*

$$\mathbb{E}\left[\sum_{t=1}^T \Delta_{K(t),t}\right] \leq (\sqrt{2}+1)\left[\left(32\kappa^2 + 4\kappa^{1.5}\sqrt{N+1}\right)\sqrt{ST\log\left(\frac{2ST}{\delta}\right)}((\Delta h)^{\max} + 1)\right]$$

$$+ \max\left(S, S\log_2\left(\frac{8T}{S}\right)\right)\left[S(\Delta h)^{\max} + (8\delta + 8S\kappa^2)\frac{q^{\max} + 1}{q^{\min}}\right]$$

*This corresponds to $\tilde{O}(\kappa^3 S^{1.5}\sqrt{T} + \kappa^{2.5}S^{1.5}\sqrt{NT})$ log-adjusted complexity bounds.*

*Proof.* First, we establish that the regret is equal to the adjusted regret in expectation

$$\mathbb{E}[\Delta_k] = \mathbb{E}[\bar{\Delta}_k] \tag{26}$$

We do this by defining another martingale difference sequence, $\Phi_{k,t}$. Let $\Delta_{k,t}$ be the per-step regret at step $t$ and $\bar{\Delta}_{k,t}$ be the per-step adjusted regret.

$$\Phi_{k,t} = \Delta_{k,t} - \bar{\Delta}_{k,t} \tag{27}$$

We can then derive, based on the independence of the event probabilities from the sojourn time, that the expected regret conditioned on a certain state $s'$ is equal to

$$\mathbb{E}[\Delta_{k,t}|s(t) = s'] = g_\pi \bar{\tau}(s') - r^h(s')\bar{\tau}(s') - \sum_{i=1}^{N} \sum_{a \in \mathcal{A}} \pi_k(s,a) 1_{i \in a_1} \bar{\tau}(s')\lambda_i(s') r_i^c(s')$$

$$- \sum_{j=1}^{M} \sum_{a \in \mathcal{A}} \pi_k(s,a) 1_{j \in a_2} \bar{\tau}\mu_j(s') r_j^s(s') - \bar{\tau}(s')\eta(s') r^a(s') - \bar{\tau}(s')\gamma(s') r^a(s')$$

$$= \mathbb{E}[\bar{\Delta}_{k,t}|s(t) = s']$$

Therefore $\mathbb{E}[\Phi_{k,t}|s(t) = s'] = 0$ for any state $s'$, and together with the Markov property this implies that $\mathbb{E}[\Phi_{k,t}|s(1), \ldots s(k-1)] = 0$. To show this is a martingale difference sequence, we must then consider the expected absolute value. Again, we condition on an arbitrary state $s'$. Noting that all rewards are bounded between $[-1,1]$, that $\bar{\tau}(s') \leq (q^{\min})^{-1}$, and all rates have appropriate upper bounds, the following holds

$$\mathbb{E}[|\Delta_{k,t}| \, |s(t) = s'] = \mathbb{E}[|\bar{\Delta}_{k,t}| \, |s(t) = s']$$

$$\leq |g_\pi| \bar{\tau}(s') + |r^h(s')| \bar{\tau}(s')$$

$$+ \sum_{i=1}^{N} \sum_{a \in \mathcal{A}} \pi_k(s,a) 1_{i \in a_1} \bar{\tau}(s')\lambda_i(s') |r_i^c(s')|$$

$$+ \sum_{j=1}^{M} \sum_{a \in \mathcal{A}} \pi_k(s,a) 1_{j \in a_2} \bar{\tau}\mu_j(s') |r_j^s(s')|$$

$$+ \bar{\tau}(s')\eta(s') |r^a(s')| + \bar{\tau}(s')\gamma(s') |r^a(s')|$$

$$< \infty$$

Therefore by the optional stopping theorem

$$\mathbb{E}\left[\sum_{t=V_{k-1}+1}^{v_k} \Phi_t\right] = 0$$

This, combined with (27) directly implies (26). Then, all that remains is to bound the total expected adjusted regret up to each step. We then use the following identity, similar to in (Auer et al., 2008)

$$\sum_k \sum_{s \in F_k^{bal}} \frac{v_k(s)}{\sqrt{V_{k-1}(s)}} \leq (\sqrt{2}+1)\sqrt{ST} \tag{28}$$

Summing over the results in lemmas 8, 9, and 12, we have

$$\mathbb{E}[\sum_k \Delta_k] = \mathbb{E}[\sum_k \bar{\Delta}_k] \leq \mathbb{E}\left[\sum_k \left(\sum_{s \in F_k^{bal}} \frac{v_k(s)}{\sqrt{V_{k-1}(s)}}\right) Y + Z\right]$$

Where

$$Y = \left[\left(32\kappa^{2.5} + 4\kappa\sqrt{N+1}\right)\sqrt{\log\left(\frac{2S}{\delta_k}\right)}((\Delta h)^{\max} + 1)\right]$$

$$Z = S(\Delta h)^{\max} + 8\frac{q^{\max}+1}{q^{\min}}\delta + 8\kappa^2 S \frac{q^{\max}+1}{q^{\min}}$$

Then, applying (28) and Lemma 7

$$\mathbb{E}[\sum_k \Delta_k] \leq (\sqrt{2}+1)\left[\left(32\kappa^2 + 4\kappa^{1.5}\sqrt{N+1}\right)\sqrt{ST\log\left(\frac{2S}{\delta_k}\right)}((\Delta h)^{\max} + 1)\right]$$

$$+ \max\left(S, S\log_2\left(\frac{8T}{S}\right)\right)\left[S(\Delta h)^{\max} + (8\delta + 8\kappa^2 S)\frac{q^{\max}+1}{q^{\min}}\right]$$

$\square$

# D  OTHER SUPPLEMENTARY MATERIAL

## D.1  HOW THE PROPOSED MODEL GENERALIZES CERTAIN QUEUEING MODELS

In this subsection, we consider a few models that are common in the queueing literature that can be represented by the model given in the paper. These are not meant to be comprehensive, and are somewhat simplified for clarity. However, it is meant to illustrate the possible applications of the model.

### D.1.1  THE M/M/K/S QUEUE

First, we show how the proposed model, even under Assumption 1, can generalize common models from the literature. We begin with admission control for the standard finite capacity, multi-server $M/M/k/S$ queue, with $N^c$ customer types with arrival rates $\lambda_i$, no servers, and a service rate $\mu$. We set $\underline{s} = 0$, $\bar{s} = S$, and then use the following parametrization for each rate, which fulfills Assumption 1.

$$\lambda_i(s) = \lambda_i$$
$$\gamma(s) = \min(s, k)\mu$$

This is because there are no assumptions other than monotonicity and lack of agent rejection in the "abandonment" rates in the model, and therefore they can be used to model more general cases. This model can be extended to the case with abandonment rates as well. For notational clarity, let $\eta$ represent the abandonment rate of customers, and then we can set

$$\gamma(s) = \min(s, k)\mu + s\eta$$

### D.1.2  SPEED-UP AND SERVER ACTIVATION REGIMES

The standard model of multiple homogeneous servers may be restrictive in practice for modeling several real-world systems. For example, a common situation is in which service speeds up as the workload increases, either through additional servers or increasing the workload on each server. For example, (Yom-Tov & Chan, 2021) features a model based on the use of additional beds or alternative care facilities in healthcare. Another example is (Bekker et al., 2011), which models increased service in call centers to ensure a low waiting time. There are two options for modeling this. The first is by using a monotonic abandonment rate $\gamma(s)$ for situations in which speed-up or scaling is out of control of the agent. Another natural option is to use server types, with appropriate rewards or costs, for modeling the choice of whether or not to add additional service when it is under control of an agent.

### D.1.3  TWO-SIDED MARKETS WITH STRATEGIC BEHAVIOR

Next, we consider a two-sided queue in which the service discipline is first-come-first-served, (FCFS), and each entity enters strategically and has full knowledge of the expected waiting time. In particular, for each customer type $i$ there exists a cost of waiting $c_i^c < 0$ incurred per unit of time, and a utility $u_i^c > 0$ for being paired with a server. For each server type, there is a cost of waiting $c_j^s < 0$ per unit of time, and a utility $u_j^s > 0$ for being paired with a customer. Customers and servers arrive to observe the queue with state-independent rates $\lambda_i$ and $\mu_j$, for a customer and server type $i$ and $j$ respectively. Customers and servers also abandon or are cleared from the queue, non-strategically, with rates $\gamma$ and $\eta$, respectively. However, they may decide not to enter the system based on the state, without the control of the agent. This bears similarity to the classical model of Naor (Naor, 1969), and several models in rational queueing with observable queues (Hassin, 2016). Based on the model, each customer requests entry to the queue in state $s \geq 0$ if

$$u_i^c \geq c_i^c s \sum_{j=1}^{N^s} \mu_j$$

In each state $s < 0$, each customer will request entry to the queue since they receive an immediate positive reward of $u_i^c$. Therefore, the arrival rate for each customer type can be modeled as

$$\lambda_i(s) = \begin{cases} \lambda_i & s \leq u_i^c \left( c_i^c \sum_{j=1}^{N^s} \mu_j \right)^{-1} \\ 0 & \text{otherwise} \end{cases}$$

With a similar argument, we can derive

$$\mu_j(s) = \begin{cases} \mu_j & s \geq u_i^s \left( c_i^c \sum_{i=1}^{N^c} \lambda_i \right)^{-1} \\ 0 & \text{otherwise} \end{cases}$$

In practical situations customers are generally not purely strategic and do not have full observability. Therefore, their reaction to increasing queue lengths and waiting times may need to be learned rather than modeled. However, in general rate monotonicity could be a reasonable a priori assumption when customers and servers incur a cost for waiting.

## D.2 SUPPLEMENTARY DEFINITIONS

Some relevant terms used in the paper from the theory of continuous-time Markov chains include sojourn times and transient states. Let $Q$ be the generator matrix of a continuous-time Markov chain. The sojourn time of state $s$ in a continuous-time Markov chain is a random variable, and is exponentially distributed with a rate equal to the hidden transition rate $-Q_{s,s}$. A transient state is a state that occurs infinitely often with a probability of $0$, and a recurrent state is a state that is not transient. In other words, the system almost surely enters and remains in a class of recurrent states given sufficient time.

## D.3 DETAILED PSEUDOCODE FOR EXTENDED MODEL CONSTRUCTION

Here, we present a more detailed pseudocode for the construction of the extended model, than was presented in the main paper. Algorithm 4 gives the core algorithm, using Algorithm 5 as a subroutine for constructing both the abandonment rates as well as the arrival rate of each type Bolch et al. (2006).

---

**Algorithm 4** Constructing the Extended Model

**Require:** Sojourn time observations $\tau_t$, event counts $v(\cdot,\cdot)$, confidence parameter $\delta$

1: **for** $s = \underline{s} \ldots \bar{s}$ **do**
2:    Set $\gamma_k^{ext}(s) = \frac{\max(\hat{p}_{k,s}^+(0) - \frac{1}{2}\varepsilon_k^{p^+}(s), 0)}{\hat{\tau}_k^+(s) + \varepsilon_k^{\tau^+}(s)}$ if $s > 0$
3:    Set $\eta_k^{ext}(s) = \frac{\max(\hat{p}_{k,s}^-(0) - \frac{1}{2}\varepsilon_k^{p^-}(s), 0)}{\hat{\tau}_k^-(s) + \varepsilon_k^{\tau^-}(s)}$ if $s < 0$
4: **end for**
5: Set $\gamma_k^{ext}(\bar{s}+1), \eta_k^{ext}(\underline{s}-1) = -\infty$
6: **for** $s = \bar{s} \ldots 1$ **do**
7:    Set $\gamma_k^{ext}(s) = \max(\gamma_k^{ext}(s), \gamma_k^{ext}(s+1), q^{\min})$
8: **end for**
9: **for** $s = \underline{s} \ldots -1$ **do**
10:    Set $\gamma_k^{ext}(s) = \max(\eta_k^{ext}(s), \eta_k^{ext}(s-1), q^{\min})$
11: **end for**
12: **for** $s = \underline{s} \ldots \bar{s}$ **do**
13:    Set $q^+(s) = (\hat{\tau}_k^+(s) - \varepsilon_k^{\tau^+}(s))^{-1}$
14:    Set $q^-(s) = (\hat{\tau}_k^-(s) - \varepsilon_k^{\tau^-}(s))^{-1}$
15: **end for**
16: Set $q^+(\underline{s}-1), q^-(\bar{s}+1) = \infty$
17: **for** $s = \underline{s} \ldots \bar{s}-1$ **do**
18:    Set $q^+(s) = \min(q^+(s), q^+(s-1), \Lambda^{\max} + 1_{s<0}\eta^{\max})$
19:    Set $q^-(s) = \min(q^+(s), q^-(s-1), M^{\max} + 1_{s>0}\gamma^{\max})$
20: **end for**
21: **for** $s = \underline{s} \ldots \bar{s}$ **do**
22:    Follow algorithm 5 to find individual arrival and abandonment rates
23: **end for**

---

---

**Algorithm 5** Finding optimistic arrival and abandonment rates

---

**Require:** State $s$

1: Initialize total rate differences $\Delta q^+(s) = q^+(s)\varepsilon^{p^+}(s)$, $\Delta q^-(s) = q^-(s)\varepsilon^{p^-}(s)$

2: **for** $l \in argsort([-N^s, N^c], r_{l_1}(s) \le r_{l_2}(s)\dots)$ **do**

3:    **if** $l = 0$ **and** $s < 0$ **and** $q^+(s)\hat{p}^+_{k,s}(0) \ge \eta^{ext}(s)$ **then**

4:       Set $\bar{\eta}^{ext}(s) = \max(\eta^{ext}(s), q^+(s)\hat{p}^+_{k,s}(0) - \Delta q^+(s))$

5:       Set $\Delta q^+(s) = \Delta q^+(s) - q^+(s)\hat{p}^+_{k,s}(0) + \bar{\eta}^{ext}(s)$

6:    **else if** $l > 0$ **then**

7:       Set $\lambda^{ext}_l(s) = \max(0, q^+(s)\hat{p}^+_{k,s}(l) - \Delta q^+(s))$

8:       Set $\Delta q^+(s) = \Delta q^+(s) - q^+(s)\hat{p}^+_{k,s}(l) + \lambda^{ext}_l(s)$

9:    **else if** $l = 0$ **and** $s > 0$ **and** $q^-(s)\hat{p}^-_{k,s}(0) \ge \eta^{ext}(s)$ **then**

10:      Set $\bar{\gamma}^{ext}(s) = \max(\gamma^{ext}(s), q^-(s)\hat{p}^-_{k,s}(0) - \Delta q^-(s))$

11:      Set $\Delta q^-(s) = \Delta q^-(s) - q^-(s)\hat{p}^-_{k,s}(0) + \bar{\gamma}^{ext}(s)$

12:    **else if** $l < 0$ **then**

13:      Set $\mu^{ext}_{-l}(s) = \max(0, q^-(s)\hat{p}^-_{k,s}(l) - \Delta q^-(s))$

14:      Set $\Delta q^+(s) = \Delta q^-(s) - q^+(s)\hat{p}^-_{k,s}(l) + \mu^{ext}_{-l}(s)$

15:    **end if**

16: **end for**

17: Re-initialize total rate differences, accounting for any unused differences due to abandonment
    bounds, $\Delta q^+(s) = q^+(s)\varepsilon^{p^+}(s) - \Delta q^+(s)$, $\Delta q^-(s) = q^-(s)\varepsilon^{p^-}(s) - \Delta q^-(s)$

18: **if** $q^+(s)\hat{p}^+_{k,s}(0) < \eta^{ext}(s)$ **then**

19:    Set $\Delta q^+(s) = \Delta q^+(s) - \eta^{ext}(s) + \bar{\eta}^{ext}(s)$

20:    Set $\bar{\eta}^{ext}(s) = \eta^{ext}(s)$

21: **end if**

22: **if** $q^-(s)\hat{p}^-_{k,s}(0) < \gamma^{ext}(s)$ **then**

23:    Set $\Delta q^-(s) = \Delta q^-(s) - \gamma^{ext}(s) + \bar{\gamma}^{ext}(s)$

24:    Set $\bar{\gamma}^{ext}(s) = \gamma^{ext}(s)$

25: **end if**

26: **for** $l \in argsort([-N^s, N^c], r_{l_1}(s) \ge r_{l_2}(s)\dots)$ **do**

27:    **if** $l = 0$ **and** $s < 0$ **then**

28:      Set $\bar{\eta}^{ext}(s) = \bar{\eta}^{ext}(s) + \Delta q^+(s)$

29:      **break**

30:    **else if** $l > 0$ **then**

31:      Set $\lambda^{ext}_l(s) = \lambda^{ext}_l(s) + \Delta q^+(s)$

32:      **break**

33:    **else if** $l = 0$ **and** $s > 0$ **then**

34:      Set $\bar{\gamma}^{ext}(s) = \bar{\gamma}^{ext}(s) + \Delta q^-(s)$

35:      **break**

36:    **else if** $l < 0$ **then**

37:      Set $\mu^{ext}_{-l}(s) = \mu^{ext}_{-l}(s) + \Delta q^-(s)$

38:      **break**

39:    **end if**

40:    Set $\lambda^{ext}_{N^c+1}(s) = \bar{\eta}^{ext}(s) - \eta^{ext}(s)$

41:    Set $\mu^{ext}_{N^s+1}(s) = \bar{\gamma}^{ext}(s) - \gamma^{ext}(s)$

42: **end for**

---

