# OpenReview forum: "Reinforcement Learning for Admission Control in Two-Sided Queueing Systems"
_ICLR.cc/2026/Conference — ICLR 2026 Conference Withdrawn Submission_

### Official Review · Reviewer_fDuV · 2025-10-23

**Soundness:** 3
**Presentation:** 2
**Contribution:** 2
**Rating:** 6
**Confidence:** 3

**Summary:**

This work considers the problem of admission control in two-sided queues. At one side of the queue, tasks arrive from customers. At the other side, tasks are consumed by available servers.  The decision-making problem involves whether to admit or reject incoming customers and servers. The proposed algorithm relies on a monotonicity assumption about the arrival rates (i.e., that they keep increasing). The algorithm is shown to have more favorable regret than existing generic algorithms. It is also evaluated empirically on a few toy problems.

**Strengths:**

S1. The paper has a very thorough theoretical treatment and is well-written.

**Weaknesses:**

W1. The clarity of the work could be improved in my opinion, as I detail in my comments below.

W2. This is subjective, but I find the setting of the paper somewhat convoluted and of limited practicality given the key monotonicity assumption. The assumption is not supported with arguments or real-world examples. Is it realistic to assume that customer rates keep decreasing while server rates keep increasing? However, the paper is clearly on the theoretical side.

**Questions:**

C1. Regarding "the state can be represented as an integer" (L92): is this true? While the number of customers / servers indeed needs to be tracked, this seems to be a lossy representation. Why are the types of customers and servers not represented in the state also? Is it the case that rewards for abandonments of existing customers / servers are type-independent? How can this be justified given acceptance rewards are type-dependent?

C2. Other works on RL for queue control worth considering citing are [1] and [2]. [1] in particular has a similar model where customers (tasks) and servers belong to different classes that influence the rewards received.

C3. Some notions related to queuing theory are not explained.  Could you clarify transient states and sojourn times?

C4. Small comments:
- $\Lambda(s)$ is used on L106 before being defined in Assumption 1
- L180: Alg 3 seems to be a much more compact version of Alg 4, do you want to refer to Alg 3 instead here?
- L528: "Ionnaidis" -> "Ioannidis"

### References

[1] Staffolani, A., Darvariu, V. A., Bellavista, P., & Musolesi, M. (2023). RLQ: Workload allocation with reinforcement learning in distributed queues. IEEE Transactions on Parallel and Distributed Systems, 34(3), 856-868.

[2] Jali, N., Qu, G., Wang, W., & Joshi, G. (2024, April). Efficient reinforcement learning for routing jobs in heterogeneous queueing systems. In International Conference on Artificial Intelligence and Statistics (pp. 4177-4185). PMLR.

---

> ### Author Response · Authors · 2025-11-17
>
> Thank you for your comments, we are revising the paper to improve clarity.
>
> ### On the validity of Assumption 1:
> Monotonicity assumptions [7-10], or intrinsically monotonic problems [11-14], are quite common in the queueing theoretic literature for state-dependence. A few examples of situations that can be modeled monotonic state-dependent rates include queues with strategic customers that are sensitive to the waiting time [11-13], systems with server activation and speed-up [9; 10], and standard multi-server queueing models. Strategic entities, in particular, are of particular relevance in two-sided markets. We are revising the paper to better clarify the applicability of this assumption.
>
> We would also like to clarify that Assumption 1 requires monotonicity, but not necessarily strict monotonicity.
>
> ### Responses to comments and questions:
> W1. Thank you, we will revise the paper to improve the clarity in line with your comments.
>
> W2. Please see the comment above.
>
> C1. Yes, rates are type-independent. This is quite standard in the admission control literature, including recent work [1-5]. Considering compatibilities and service differentiation would be promising for future work, but would require a different focus due to the exponential size of the state space over the size of the buffer. Furthermore, while compatibilities are a promising direction for future work, queueing systems with compatibilities only have known product-form solutions under stringent assumptions [6].
>
> C2. Thank you, we will make sure to cite these in the paper.
>
> C3. We will ensure that these are explicitly defined in the paper upon revision. Transient states are states outside the recurrent class, and therefore are only seen a finite number of times. The sojourn time is a random variable that represents the amount of time the system spends in a particular state, and is exponentially distributed with a rate equal to the sum of all transition rates.
>
> ### References
>
> [1] Miller, Bruce L. "A queueing reward system with several customer classes." Management science 16.3 (1969): 234-245.
>
> [2] Yan Su and Junping Li. Admission Control of Double-Sided Queues With Multiple Customer Types.
> IEEE Transactions on Automatic Control, 69(3):1960–1966, 2024.
>
> [3] Feinberg, Eugene A., and Fenghsu Yang. "Optimality of trunk reservation for an M/M/k/N queue with several customer types and holding costs." Probability in the Engineering and Informational Sciences 25.4 (2011): 537-560.
>
> [4] Su, Yan, and Junping Li. "Bias optimality of admission control in a non-stationary repairable queue." Operations Research Letters 48.3 (2020): 317-322.
>
> [5] Feinberg, Eugene A., and Martin I. Reiman. "Optimality of randomized trunk reservation." Probability in the Engineering and Informational Sciences 8.4 (1994): 463-489.
>
> [6] Berezner, S. A., and Anthony E. Krzesinski. "Order independent loss queues." Queueing Systems 23.1 (1996): 331-335.
> [7] Bekker, Rene. "Queues with state-dependent rates." (2005).
>
> [8] Mandelbaum, Avi, and Gennady Pats. "State-dependent stochastic networks. Part I. Approximations and applications with continuous diffusion limits." The Annals of Applied Probability 8.2 (1998): 569-646.
>
> [9] Yom-Tov, Galit B., and Carri W. Chan. "Balancing admission control, speedup, and waiting in service systems." Queueing systems 97.1 (2021): 163-219.
>
> [10] Bekker, René, et al. "Queues with waiting time dependent service." Queueing systems 68.1 (2011): 61-78.
>
> [11] Hassin, Refael. Rational queueing. CRC press, 2016.
>
> [12] Naor, Pinhas. "The regulation of queue size by levying tolls." Econometrica: journal of the Econometric Society (1969): 15-24.
>
> [13] Philipp Afeche, Zhe Liu, and Costis Maglaras. Ride-hailing networks with strategic drivers: The
> impact of platform control capabilities on performance. Manufacturing & Service Operations
> Management, 25(5):1890–1908, 2023.
>
> [14] Altman, Eitan, et al. "State-dependent M/G/1 type queueing analysis for congestion control in data networks." Proceedings IEEE INFOCOM 2001. Conference on Computer Communications. Twentieth Annual Joint Conference of the IEEE Computer and Communications Society (Cat. No. 01CH37213). Vol. 3. IEEE, 2001.

---

### Official Review · Reviewer_ysxy · 2025-10-25

**Soundness:** 2
**Presentation:** 2
**Contribution:** 2
**Rating:** 2
**Confidence:** 1

**Summary:**

This paper studies admission control in two-sided queueing systems. To address the exponential dependence on the MDP diameter that arises in general-purpose reinforcement learning algorithms, the authors propose a method with a diameter-independent regret bound. They derive a regret of $\tilde{O}(S^{1.5}\sqrt{T})$ and demonstrate the empirical performance of their algorithm using synthetic datasets.

**Strengths:**

This paper is the first to propose a reinforcement learning algorithm for two-sided queues with state-dependent arrival and service rates.

**Weaknesses:**

1. The novelty of the proposed algorithm is not clearly articulated. The method appears to rely heavily on UCRL-based techniques. Additionally, it remains unclear why incorporating state-dependent arrival and service rates presents a fundamental challenge beyond existing approaches.

2. Assumption 1 appears to be quite strong, yet no justification is provided to support its validity in practical systems. A more detailed discussion on when such monotonicity assumptions hold in real-world queueing scenarios would be valuable.

**Questions:**

1. The paper would benefit from a clearer explanation of why handling state-dependent arrival and service rates poses additional challenges compared to existing approaches.

2. It would be helpful if the paper could elaborate more clearly on the differences from UCRL-AC (Weber et al., 2024).

3. Assumption 1 is quite strong and currently lacks justification. Could the authors provide theoretical or empirical support for why such monotonicity conditions are expected to hold in real-world two-sided queueing systems? Moreover, it remains unclear why the algorithm relies on Assumption 2 instead of directly using Assumption 1; further clarification of the relationship and necessity of these assumptions would improve the paper.

\textbf{Minor comment:} $\Lambda(s)$ and $M(s)$ on page 2 should be defined before they are referenced.

---

> ### Author Response · Authors · 2025-11-17
>
> Thank you for the comments. We will make an appropriate revision to improve the clarity on both the method as well as its novelty.
>
> With regards to the novelty of admission control learning in state-dependent queues, we were unable to find existing work on learning for queue control with attractive regret guarantees that accounts for state-dependence. Please see [1-4] for related or highly-cited papers that only learn global scalar rates.
>
> ### On the validity of Assumption 1:
> Monotonicity assumptions [8-9], or intrinsically monotonic problems [10-13], are quite common in the queueing theoretic literature for state-dependence. A few examples of situations that can be modeled monotonic state-dependent rates include queues with strategic customers that are sensitive to the waiting time [10-12], systems with server activation and speed-up [8; 9], and standard multi-server queueing models. Strategic entities, in particular, are of particular relevance in two-sided markets. We are revising the paper to better clarify the applicability of this assumption.
>
> Proposition 2 (L366) shows, through use of a counterexample, that some assumption on the rates is necessary to ensure polynomial bounds on the bias. We felt that monotonicity is a well-motivated and common condition that can apply to a significant number of real-world applications.
>
>
> ### Responses to Questions
> 1. With regards to the difficulty of learning state-dependent rates, it requires more exploration from the agent than learning scalar arrival/service rates. More specifically, we consider it a relatively surprising result that the impact on regret is relatively modest (i.e. still polynomial), despite having to explore a wider variety of states, each of which has an exponential hitting time. It also implies that the low regret despite the diameter in queue control problems [2], is not necessarily a result of parameter sharing between different states but is entailed by time-reversibility. Thank you for identifying this, we will revise the paper to make this contribution more explicit.
> 2. With regards to the differences with [1], there are several more differences in addition to the core contributions of considering a two-sided queue and state-dependent rates. In particular, all rates are assumed to be unknown in this paper, unlike [1] which assumes a known service rate. Furthermore, the rewards are unconstrained, and can be positive or negative for both customers, servers, and abandonments. In our opinion, this is a significant generalization from prior work.
> 3. See our comments above.
>
> ### References
>
> [1] Lucas Weber, Ana Busic, and Jiamin Zhu. Reinforcement learning and regret bounds for admission
> control. In ICML’24: Proceedings of the 41st International Conference on Machine Learning,
> pp. 52403 – 52427, 2024.
>
> [2] Jonatha Anselmi, Bruno Gaujal, and Louis-Sebastien Rebuffi. Reinforcement learning in a birth and
> death process: breaking the dependence on the state space. In Advances in neural information
> processing systems, volume 35, pp. 14464–14474, 2022.
>
> [3] Raeis, Majid, Ali Tizghadam, and Alberto Leon-Garcia. "Queue-learning: A reinforcement learning approach for providing quality of service." Proceedings of the AAAI Conference on Artificial Intelligence. Vol. 35. No. 1. 2021.
>
> [4] Stahlbuhk, Thomas, Brooke Shrader, and Eytan Modiano. "Learning algorithms for minimizing queue length regret." IEEE Transactions on Information Theory 67.3 (2021): 1759-1781.
>
> [5] Yang, Zixian, Sushil Mahavir Varma, and Lei Ying. "Near-Optimal Regret-Queue Length Tradeoff in Online Learning for Two-Sided Markets." arXiv preprint arXiv:2510.14097 (2025).
>
> [6] Bekker, Rene. "Queues with state-dependent rates." (2005).
>
> [7] Mandelbaum, Avi, and Gennady Pats. "State-dependent stochastic networks. Part I. Approximations and applications with continuous diffusion limits." The Annals of Applied Probability 8.2 (1998): 569-646.
>
> [8] Yom-Tov, Galit B., and Carri W. Chan. "Balancing admission control, speedup, and waiting in service systems." Queueing systems 97.1 (2021): 163-219.
>
> [9] Bekker, René, et al. "Queues with waiting time dependent service." Queueing systems 68.1 (2011): 61-78.
>
> [10] Hassin, Refael. Rational queueing. CRC press, 2016.
>
> [11] Naor, Pinhas. "The regulation of queue size by levying tolls." Econometrica: journal of the Econometric Society (1969): 15-24.
>
> [12] Philipp Afeche, Zhe Liu, and Costis Maglaras. Ride-hailing networks with strategic drivers: The
> impact of platform control capabilities on performance. Manufacturing & Service Operations
> Management, 25(5):1890–1908, 2023.
>
> [13] Altman, Eitan, et al. "State-dependent M/G/1 type queueing analysis for congestion control in data networks." Proceedings IEEE INFOCOM 2001. Conference on Computer Communications. Twentieth Annual Joint Conference of the IEEE Computer and Communications Society (Cat. No. 01CH37213). Vol. 3. IEEE, 2001.

---

### Official Review · Reviewer_kkDT · 2025-10-31

**Soundness:** 3
**Presentation:** 2
**Contribution:** 3
**Rating:** 4
**Confidence:** 3

**Summary:**

This paper proposes UCRL-TSAC, a reinforcement learning algorithm for admission control in two-sided queueing systems, where both customers and servers arrive stochastically and must be matched. General-purpose RL algorithms such as UCRL2 suffer from regret bounds that depend on the MDP diameter, which grows exponentially with the number of states in queueing models. The authors develop a model-based diameter-independent algorithm.

**Strengths:**

- Novel diameter-independent regret bound for MBRL for two-sided queueing.

- Provides rigorous bias analysis showing linear dependence on state space under mild monotonicity assumptions.

**Weaknesses:**

- I have general question regarding the problem formulation. Firstly, while the customer has types, the state is only the queue length which does not distinguish the customer types. So is the abandonment rate and also there is no compatibility issue between different types of customers and servers. The type only comes into play in terms of the reward.  I guess all of these are intentional to make the problem simple enough, but I wonder whether any real-world problems can fit into such a formulation, and whether the model is an important one in the queueing literature.

- I am very confused by the action and the reduced action in Line 121.  The original action is defined as all possible subsets of types, but I think this is an artifact of how the problem is formulated as the state does not contain any information of the type. Therefore, the action had to prescribe the set of types it would admit. If I were to formulate the problem, I’d make the state a tuple (q, TYPE) where q is the queue length, and TYPE is set to be the type of the arrived customer/server on the event of customer/server arrival. On the event of abandonment, TYPE is set as some special value. The action space would then just be a binary action of whether to admit or not. This way, the policy just needs to look at TYPE to know what is the event (arrival of a specific type, or abandonment) and decide whether to admit or now. The current action space in the paper is very confusing to me and I do not fully understand the restriction in Line 121. For line 126 it reads “… there exists a particular type i …” but in the equation in Line 121, it is “\forall i”

**Questions:**

See weakness

---

> ### Author Response · Authors · 2025-11-17
>
> Thank you for your comments. We will revise the paper accordingly to improve the clarity of the method and its relationship to the literature.
>
> ### On the model formulation
> Both the model formulation, and the use of types in an otherwise single-class model, are quite standard in the admission control literature [1;3;5], including recent papers [2;4;6]. We will revise the paper to ensure that the formulation is made clear.
>
> ### Clarification on the reduced action space:
> We agree that there is a slight lack of clarity on what i and j mean in the paragraph following the equation on L121, and will improve this passage. In particular, each action within the restricted action set (besides the null set) has an admittance set in which each reward-index pair is greater (lexicographically) than that of a certain action-specific threshold. The $\forall$ quantifier is used since every type has a corresponding entry in the restricted action set.
>
> ### On the validity of Assumption 1:
> Monotonicity assumptions [7-10], or intrinsically monotonic problems [11-14], are quite common in the queueing theoretic literature for state-dependence. A few examples of situations that can be modeled monotonic state-dependent rates include queues with strategic customers that are sensitive to the waiting time [11-13], systems with server activation and speed-up [9; 10], and standard multi-server queueing models. Strategic entities, in particular, are of particular relevance in two-sided markets. We are revising the paper to better clarify the applicability of this assumption.
>
> Proposition 2 (L366) shows, through use of a counterexample, that some assumption on the rates is necessary to ensure polynomial bounds on the bias. We felt that monotonicity is a well-motivated and common condition that can apply to a significant number of real-world applications.
>
> We would also like to clarify that Assumption 1 requires monotonicity, but not necessarily strict monotonicity.
>
> ### References
>
> [1] Miller, Bruce L. "A queueing reward system with several customer classes." Management science 16.3 (1969): 234-245.
>
> [2] Yan Su and Junping Li. Admission Control of Double-Sided Queues With Multiple Customer Types.
> IEEE Transactions on Automatic Control, 69(3):1960–1966, 2024.
>
> [3] Feinberg, Eugene A., and Fenghsu Yang. "Optimality of trunk reservation for an M/M/k/N queue with several customer types and holding costs." Probability in the Engineering and Informational Sciences 25.4 (2011): 537-560.
>
> Feinberg, Eugene A., and Fenghsu Yang. "Optimal admission to an M/M/k/N queue with several customer types and holding costs." 49th IEEE Conference on Decision and Control (CDC). IEEE, 2010.
>
> [4] Su, Yan, and Junping Li. "Bias optimality of admission control in a non-stationary repairable queue." Operations Research Letters 48.3 (2020): 317-322.
>
> [5] Feinberg, Eugene A., and Martin I. Reiman. "Optimality of randomized trunk reservation." Probability in the Engineering and Informational Sciences 8.4 (1994): 463-489.
>
> [6] Lucas Weber, Ana Busic, and Jiamin Zhu. Reinforcement learning and regret bounds for admission
> control. In ICML’24: Proceedings of the 41st International Conference on Machine Learning,
> pp. 52403 – 52427, 2024.
>
> [7] Bekker, Rene. "Queues with state-dependent rates." (2005).
>
> [8] Mandelbaum, Avi, and Gennady Pats. "State-dependent stochastic networks. Part I. Approximations and applications with continuous diffusion limits." The Annals of Applied Probability 8.2 (1998): 569-646.
>
> [9] Yom-Tov, Galit B., and Carri W. Chan. "Balancing admission control, speedup, and waiting in service systems." Queueing systems 97.1 (2021): 163-219.
>
> [10] Bekker, René, et al. "Queues with waiting time dependent service." Queueing systems 68.1 (2011): 61-78.
>
> [11] Hassin, Refael. Rational queueing. CRC press, 2016.
>
> [12] Naor, Pinhas. "The regulation of queue size by levying tolls." Econometrica: journal of the Econometric Society (1969): 15-24.
>
> [13] Philipp Afeche, Zhe Liu, and Costis Maglaras. Ride-hailing networks with strategic drivers: The
> impact of platform control capabilities on performance. Manufacturing & Service Operations
> Management, 25(5):1890–1908, 2023.
>
> [14] Altman, Eitan, et al. "State-dependent M/G/1 type queueing analysis for congestion control in data networks." Proceedings IEEE INFOCOM 2001. Conference on Computer Communications. Twentieth Annual Joint Conference of the IEEE Computer and Communications Society (Cat. No. 01CH37213). Vol. 3. IEEE, 2001.

---

### Official Review · Reviewer_cXjD · 2025-11-06

**Soundness:** 2
**Presentation:** 1
**Contribution:** 2
**Rating:** 2
**Confidence:** 4

**Summary:**

The authors consider the usage of reinforcement learning for admission control in two sided queuing networks. The approach is model based, that is an UCRL type of analysis - where you first build an estimate of the MDP and then solve for the optimal policy. The state space consists of net number of customers or servers. The arrival rates for both types are allowed to be state dependent. The abandonment rates for both types are also allowed to be state dependent. The servers are _assumed_ to be arriving at a higher rate as the number of customers increase and vice versa. The action space considers a restricted set of actions. The main contribution is to arrive as regret bounds which do not scale with the diameter. The authors also perform some experimental validation.

**Strengths:**

1. Considers an average reward approach for this problem, which is technically challenging but better captures the objective.
2. Provides diameter independent bounds with state dependent rates of arrival and departure.
3. Considers a reduced action space with might help with computational complexity issues.

**Weaknesses:**

1. The paper needs to be better written. The model (MDP) is not specified clearly. Notations are used much before they are defined. Not all assumptions are in the main body.
2. Literature survey is insufficient. There has been quite some development in this domain which is very relevant to the topic studied in this paper that has not been discussed. I am adding a few of these references below.
3. The authors mention that other works assume the stability of policies. Since the problem considered in this paper assumes a finite state space, stability is an inherent assumption of the work.
4. These problems typically fall in the regime of countable state space systems. That is, most applications have incredibly large state spaces, that a model based approach where the transition matrix is explicitly constructed being the right way to solve for this problem is unclear. Some of the references below handle countable state space systems, making their guarantees meaningful in large scale systems.
5. The modeling assumptions are very strong. For example the abandonment rates are very specific and dont always the reality. The arrival rates are also subject to strong assumptions. The overall setting considered to be quite stylized and not quite general.
6. Even though its the RL setting, the bounds on these arrival rates and abandonment rates are assumed to be known.
7. The constant $\kappa$ can scale exponentially with the state space size making the lack of diameter dependence sort of moot.

It will help if the authors can better contrast their work with the rest of the work in literature and identify the exact contributions, as that is very unclear as of now.




Refs:
1. Score-Aware Policy-Gradient and Performance Guarantees using Local Lyapunov Stability Comte et al.
2. Queueing network controls via deep reinforcement learning Dai and Gluzman
3. Performance of npg in countable state-space average-cost rl Murthy et al
4. Near-Optimal Regret-Queue Length Tradeoff in Online Learning for Two-Sided Markets Yang et al.

**Questions:**

See above.

---

> ### Author Response · Authors · 2025-11-17
> **Response (1/2)**
>
> Thank you for the comments. We will take these into account when revising the paper.
>
> ### On the novelty and contributions of the paper:
> The core of the impact is about learning for state-dependent queues. This case is seldom considered in the literature. For example, [1-5] are related or highly-cited papers on learning for queue control, and none feature state-dependent rates. [6] does allow for some state-dependence under the assumption that the queue is a flow-equivalent server of a Jackson network, but assumes constant arrival rates, and it is still strictly more restrictive than the environment given in the paper. [7], which is one of the papers suggested in the review, also assumes that the arrival rate of any class is independent of the state. We have not seen any papers that establish attractive regret guarantees for queueing problems with more general state dependence and thus we believe this is a novel contribution.
>
> Achieving square root regret guarantees with polynomial scaling over the state-space with state-dependent rates is much more surprising than the structured results in the literature, as the algorithm must explore much more to learn the rates at arbitrary states. Since these states may have exponential hitting times, this is a rather unexpected result.
>
> Also, we note many of the identified countable-space models have worse regret guarantees with regards to T than what we have presented, but that is partially intrinsic to policy-gradient methods.
>
>
> ### On the use of abandonments:
> The use of the term abandonment is a convenient term for the model. It was mentioned at the end of Section II (L160-161) that this can be used to model service rates in one-sided queues. The proposed model generalizes service in one-sided queues in two ways:
>
> 1. Server arrivals, which may be accepted or rejected by the agent.
>
> 2. A mandatory service process, which we term abandonment to cleanly distinguish it from above.
>
> ### On the validity of Assumption 1:
> Monotonicity assumptions [13-16], or intrinsically monotonic problems [17-19], are quite common in the queueing theoretic literature for state-dependence. A few examples of situations that can be modeled monotonic state-dependent rates include queues with strategic customers that are sensitive to the waiting time [17-19], systems with server activation and speed-up [15; 16], and standard multi-server queueing models. Strategic entities, in particular, are of particular relevance in two-sided markets. We are revising the paper to better clarify the applicability of this assumption.
>
> Proposition 2 (L366) shows, through use of a counterexample, that some assumption on the rates is necessary to ensure polynomial bounds on the bias. We felt that monotonicity is a well-motivated and common condition that can apply to a significant number of real-world applications.
>
> We would also like to clarify that Assumption 1 requires monotonicity, but not necessarily strict monotonicity.
>
> ### On countability:
> Finite state space models, including closed queueing networks, loss networks, and finite-capacity queues are very common and remain an active area of research. Furthermore, much of the motivation for using countable state space systems is analytic tractability, not just fidelity at scale. Regarding stability, two-sided queues are not positive recurrent with infinite buffers [20], unless there are stronger assumptions such as a strictly increasing abandonment process.
>
> We also note that we consider the admission control problem in this paper. If there is a holding cost, then the system will have a finite number of recurrent states under the optimal policy. Therefore, a countable state model won't significantly improve the generality of the proposed algorithm.
>
> Since the model is product-form under any policy, the gain and bias can be found quickly without needing to solve the gain-bias equations simultaneously. Proposition 5 (L637) and Proposition 6 (L655) can be used to find the state probabilities and bias under any state, and therefore policy improvement is possible in O(S) time.

---

> ### Author Response · Authors · 2025-11-17
> **Response (2/2)**
>
> ### About the identified weaknesses:
> 1. Thank you for letting us know, we will fix the notational issues. We note that the Assumptions given in the Appendix are weaker and strictly implied by those given in the main paper for the true model. These are listed separately since they are fulfilled by the optimistic model, which may violate Assumption 1 (see L191-197, L596-601).
> 2. Thank you for these references, we will expand the literature review accordingly.
> 3. We meant that [6] requires that the traffic intensity is less than 1 at every queue in the network, which is a rather restrictive assumption for the admission control problem, even in finite-state scenarios. This point will be better clarified in the revision.
> 4. See our comment above about countability.
> 5. See our comment above for a defense on the generality and usefulness of the monotonicity assumption.
> 6. This is reasonably standard in the literature on model-based continuous-time reinforcement learning [1;8;9]. We have not seen confidence intervals with finite-sample guarantees for the rate of an exponential distribution that do not require a priori bounds due to the heavy tail, for example all of the intervals proposed in [10] require a priori bounds. We do not feel that this is any more severe than having a priori bounds on the rewards in UCRL2 or its variations.
> 7. We would like to clarify that $\kappa$ is defined as the ratio between the maximum and minimum bounds for the total transition rate at any state (see L109 and the abstract), and does not scale with the number of states.
>
> ### References
> [1] Lucas Weber, Ana Busic, and Jiamin Zhu. Reinforcement learning and regret bounds for admission
> control. In ICML’24: Proceedings of the 41st International Conference on Machine Learning,
> pp. 52403 – 52427, 2024.
>
> [2] Jonatha Anselmi, Bruno Gaujal, and Louis-Sebastien Rebuffi. Reinforcement learning in a birth and
> death process: breaking the dependence on the state space. In Advances in neural information
> processing systems, volume 35, pp. 14464–14474, 2022.
>
> [3] Raeis, Majid, Ali Tizghadam, and Alberto Leon-Garcia. "Queue-learning: A reinforcement learning approach for providing quality of service." Proceedings of the AAAI Conference on Artificial Intelligence. Vol. 35. No. 1. 2021.
>
> [4] Freund, Daniel, Thodoris Lykouris, and Wentao Weng. "Quantifying the cost of learning in queueing systems." Advances in Neural Information Processing Systems 36 (2023): 6532-6544.
>
> [5] Stahlbuhk, Thomas, Brooke Shrader, and Eytan Modiano. "Learning algorithms for minimizing queue length regret." IEEE Transactions on Information Theory 67.3 (2021): 1759-1781.
>
> [6] Anselmi, Jonatha, Bruno Gaujal, and Louis-Sébastien Rebuffi. "Learning optimal admission control in partially observable queueing networks." Queueing Systems 108.1 (2024): 31-79.
>
> [7] Yang, Zixian, Sushil Mahavir Varma, and Lei Ying. "Near-Optimal Regret-Queue Length Tradeoff in Online Learning for Two-Sided Markets." arXiv preprint arXiv:2510.14097 (2025).
>
> [8] Gao, Xuefeng, and Xun Yu Zhou. "Logarithmic regret bounds for continuous-time average-reward Markov decision processes." SIAM Journal on Control and Optimization 62.5 (2024): 2529-2556.
>
> [9] Gao, Xuefeng, and Xunyu Zhou. "Square-root regret bounds for continuous-time episodic markov decision processes." Mathematics of Operations Research (2025).
>
> [10] Bubeck, Sébastien, Nicolo Cesa-Bianchi, and Gábor Lugosi. "Bandits with heavy tail." IEEE Transactions on Information Theory 59.11 (2013): 7711-7717.
>
> [11] Comte, Céline, et al. "Score-Aware Policy-Gradient and Performance Guarantees using Local Lyapunov Stability." Journal of Machine Learning Research 26.132 (2025): 1-74.
>
> [12] Murthy, Yashaswini, et al. "Performance of npg in countable state-space average-cost rl." arXiv preprint arXiv:2405.20467 (2024).
>
> [13] Bekker, Rene. "Queues with state-dependent rates." (2005).
>
> [14] Mandelbaum, Avi, and Gennady Pats. "State-dependent stochastic networks. Part I. Approximations and applications with continuous diffusion limits." The Annals of Applied Probability 8.2 (1998): 569-646.
>
> [15] Yom-Tov, Galit B., and Carri W. Chan. "Balancing admission control, speedup, and waiting in service systems." Queueing systems 97.1 (2021): 163-219.
>
> [16] Bekker, René, et al. "Queues with waiting time dependent service." Queueing systems 68.1 (2011): 61-78.
>
> [17] Hassin, Refael. Rational queueing. CRC press, 2016.
>
> [18] Naor, Pinhas. "The regulation of queue size by levying tolls." Econometrica: journal of the Econometric Society (1969): 15-24.
>
> [19] Philipp Afeche, Zhe Liu, and Costis Maglaras. Ride-hailing networks with strategic drivers: The
> impact of platform control capabilities on performance. Manufacturing & Service Operations
> Management, 25(5):1890–1908, 2023.
>
> [20] Comte, Céline. "Stochastic non-bipartite matching models and order-independent loss queues." Stochastic Models 38.1 (2022): 1-36.

---

### Author Response · Authors · 2025-11-24
**Rebuttal Revision**

We thank the reviewers for their constructive comments, and have completed a revision of the paper accordingly. In particular,
1. We have added a contributions section to better explain both the contributions and novelty of this work (as suggested by cXjD and ysxy).
2. We have expanded the literature review to account for recent work (as suggested by cXjD and fDuV).
3. We have added in a new subsection to the appendix, Section D.1, which explicitly shows how the proposed model, even under Assumption 1, has applications to a variety of domains and generalizes several existing models in the literature (to address concerns from cXjD, ysxy and fDuV).
4. Another new subsection, Section D.2, gives definitions related to CTMDPs that readers may be unfamiliar with (as suggested by fDuV).
5. We have updated the model introduction for greater clarity on the notation used as well as the CTMDP we consider (to address concerns from cXjD and kkDT).
6. There is a small correction to one constant used in Proposition 5, in a non-dominant term. This does not impact the main contributions or results of the paper.
7. We have included Propositions 1 and 2 (formerly Propositions 5 and 6) in the main text, which can be used to find the gain and bias efficiently without needing to explicitly construct the entire generator matrix (to address concerns from cXjD).

Relevant changes are given in blue text. We hope that this improves the paper in line with the reviewers' concerns.

---

### Note · Authors · 2025-11-28

I have read and agree with the venue's withdrawal policy on behalf of myself and my co-authors.